# Reward Redistribution for CVaR MDPs using a Bellman Operator on L-infinity

**Aneri Muni** [1 2]  **Vincent Taboga** [1 2]  **Esther Derman** [2]  **Pierre-Luc Bacon** [1 2]  **Erick Delage** [2 3]

## Abstract

Tail-end risk measures such as static conditional value-at-risk (CVaR) are used in safety-critical applications to prevent rare, yet catastrophic events. Unlike risk-neutral objectives, the static CVaR of the return depends on entire trajectories without admitting a recursive Bellman decomposition in the underlying Markov decision process. A classical resolution relies on state augmentation with a continuous variable. However, unless restricted to a specialized class of admissible value functions, this formulation induces sparse rewards and degenerate fixed points. In this work, we propose a novel formulation of the static CVaR objective based on augmentation. Our alternative approach leads to a Bellman operator with: (1) dense per-step rewards; (2) contracting properties on the full space of bounded value functions. Building on this theoretical foundation, we develop risk-averse value iteration and model-free Q-learning algorithms that rely on discretized augmented states. We further provide convergence guarantees and approximation error bounds due to discretization. Empirical results demonstrate that our algorithms successfully learn CVaR-sensitive policies and achieve effective performance-safety trade-offs. Code for the paper is available at: https://github.com/amuni3/StaticCVaR-DP.

## 1. Introduction

Classical reinforcement learning (RL) theory focuses on maximizing the *expected* return over a finite or infinite decision horizon. This objective is convenient for its recursively decomposable structure, which guarantees the existence of an optimal stationary policy (Puterman, 1994). This underlying Markov recursion is the algorithmic backbone

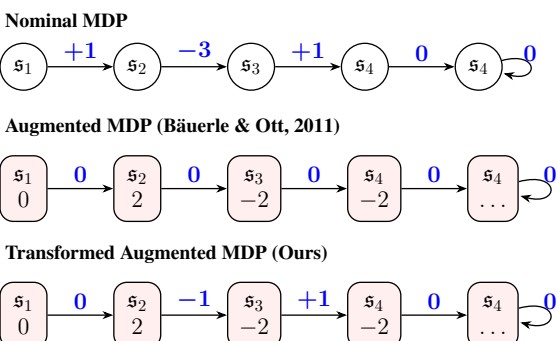

*Figure 1.* Illustration of the reward schemes produced along a trajectory under the augmented MDP of Bäuerle & Ott (2011) versus ours, highlighting per-step rewards in blue and augmented states in pink. (Refer to Appendix A for further details.)

of dynamic programming (DP), temporal-difference (TD) learning, and modern deep RL algorithms.

Despite undeniable success, average-performance-based decision-making may not always be desired. In safety-critical applications where failures might be rare yet catastrophic, a policy can be optimal in expectation yet operationally unacceptable. For example, an autonomous car that may improve its mean travel time by introducing a small collision probability or a clinical treatment plan that may improve expected outcomes, yet occasionally lead to higher morbidity in a minority subpopulation. In such settings, one would reasonably prefer to explicitly control the lower tail of the outcome distribution rather than optimize its average. Risk-sensitive RL captures this objective by replacing the expectation with a risk functional of the return distribution, thus optimizing policies with respect to tail-relevant statistics rather than mean (Wang & Chapman, 2022).

Risk-sensitive objectives can either be modeled as *nested* (recursive) risks, by applying the risk measure to the return, at every step as the process unfolds (Ruszczyński, 2010; Tamar et al., 2015a; Coache & Jaimungal, 2024), or as *static* risks, where a risk functional is applied to the full-horizon return distribution (Bäuerle & Ott, 2011; Tamar et al., 2015b; Moghimi & Ku, 2025). The nested formulation gives a convenient Markov decomposition, but need not coincide with a measure of the lower tail of the overall return distribution. The resulting nested risk-averse policies tend to be more difficult to interpret (Majumdar & Pavone,

[1]Université de Montréal, Quebec, Canada [2]Mila - Quebec AI Institute, Quebec, Canada [3]GERAD & Department of Decision Sciences, HEC Montréal, Quebec, Canada. Correspondence to: Aneri Muni <aneri.muni@mila.quebec>.

*Proceedings of the 43rd International Conference on Machine Learning*, Seoul, South Korea. PMLR 306, 2026. Copyright 2026 by the author(s).

2020) and often lead to overly conservative behavior (Lim & Malik, 2022). Static risks, in contrast, directly quantify the tail risk of cumulative returns, aligning more naturally with how risk is understood in practice. In this paper, we specifically focus on optimizing static conditional value-at-risk (CVaR), a widely used functional that holds desirable coherence properties (Artzner et al., 1999).

As with other static risk measures, the static CVaR objective lacks recursive separability and generally cannot be expressed as a Markovian Bellman recursion over the nominal state alone. This in turn leads to history-dependent optimal policies that may be intractable as the decision horizon grows (Shapiro, 2009). In their influential work, Bäuerle & Ott (2011) showed that decision-making under static CVaR can nevertheless be solved via dynamic programming, if we augment the state space with a continuous variable that tracks the return accumulated over trajectories. This augmented state, also called "budget", is a *sufficient statistic* for static CVaR and as a result, one can find policies in the augmented state space that are both stationary and optimal.

While analytically appealing, the Bellman formulation used in Bäuerle & Ott (2011) has important practical limitations. First, it concentrates all CVaR-relevant information into a terminal functional of the augmented state, thus inducing a sparse learning signal. As a consequence, this construction creates delayed feedback, even when the underlying environment provides informative per-stage rewards (see Figure 1). Sparse-rewards exacerbate exploration and temporal credit assignment challenges, becoming particularly brittle in the infinite-horizon setting (Minsky, 2007; Pignatelli et al., 2024). Second, the valid convergence properties of the associated Bellman operator are only guaranteed on a carefully restricted set of admissible unbounded functions. In practice, this means that value iteration provably converges to the right fixed point only when initialized inside this very restricted set. It is unclear how to enforce such a requirement when using function approximation or trying to deploy Q-learning methods. Alternatively, one can show that employing the same operator on space of bounded functions leads to a degenerate fixed point.

Our main contribution is to show that these limitations are not intrinsic to the static CVaR formulation, but rather an artifact of the structure of the value function it identifies. Building on Bäuerle & Ott (2011), we propose a simple algebraic transformation of static CVaR from which we derive a transformed Bellman operator and an *equivalent* fixed point. Our reformulation produces: (1) redistributed CVaR-relevant signals across time, easing integration with DP and TD-learning (Figure 1); (2) a valid contracting Bellman operator on the space of bounded functions, which enable discretization-based and model-free solution schemes.

We tackle the continuous augmented state using discretiza-

tion, based on which tractable Bellman operators can be derived. They approximate the original operator in the form of lower and upper bounds. The performance gaps we establish further highlight a proportional dependence of the error with the granularity level and the discount factor.

The contributions of this paper can be described as follows:

- A new Bellman operator for the *infinite-horizon* static CVaR objective that (i) operates on an augmented state space in the spirit of Bäuerle & Ott (2011), (ii) is exact with respect to the desired static CVaR control objective, (iii) provides a non-sparse, informative reward signal at each stage and, (iv) is a contraction mapping on the space of bounded functions.

- To overcome the computational challenge of handling a continuous augmented state, we combine uniform discretization with approximate DP and propose risk-aware value iteration and Q-learning algorithms. We further provide approximation error bounds as a function of the discretization resolution.

- Our algorithms learn optimal policies and value functions over the full augmented-state grid, yielding globally optimal solutions for all risk levels (i.e., all initial augmented states) in a single run. We also establish sufficient conditions for Q-learning convergence and validate the theory empirically.

## 2. Related Works

Several works have tried to tackle the *static* risk-sensitive RL problem. A foundational work is that of Bäuerle & Ott (2011), who proposed a state-augmentation approach to find optimal risk-averse policies for static CVaR. This work has led to several follow-up papers e.g., (Bisi et al., 2022; Lim & Malik, 2022; Bastani et al., 2022; Wang et al., 2023; Ni et al., 2024; Ávila Pires et al., 2025).

A different line of work attempts to recursively decompose CVaR into a time-consistent risk criterion. The authors of Chow et al. (2015); Stanko & Macek (2019) leverage the temporal decomposition of the dual CVaR representation, proposed by Pflug & Pichler (2016), to design DP-style algorithms for the static CVaR problem. However, recently Hau et al. (2023); Godbout & Durand (2025) showed that this decomposition fails to optimize the right objective and does not recover the true optimal policy. A different approach by Haskell & Jain (2015) relies on discretizing the occupancy measures in the augmented MDP to formulate DP algorithms for static CVaR. This method is computationally heavy, and requires solving a non-convex program using a sequence of linear-programming approximations. Recently, the authors of Kim & Min (2026) propose a Bellman operator based on the CVaR decomposition of Kim & Min

(2024), using two predictive functions (tail value and tail probability) to recover a Bellman-style recursion for static CVaR. Apart from the added computational complexity of maintaining two function approximators, their approach is limited to the finite-horizon setting.

A parallel line of work is that of Distributional RL (Dabney et al., 2018b;a; Bellemare et al., 2023), that aims to learn the entire return distribution from which a risk-neutral policy can be derived. Several works have leveraged this framework to design algorithms for risk-sensitive RL (Dabney et al., 2018a;b; Bodnar et al., 2020; Keramati et al., 2020; Schneider et al., 2024), however Lim & Malik (2022) showed that these methods learn a policy that is optimal for neither the nested risk nor for static risk objectives. Bäuerle et al. (2026) recently proposed a new distributional Bellman operator for static risks, however it inherits the sparsity issues from Bäuerle & Ott (2011). Ávila Pires et al. (2025) also proposed a distributional DP approach that solves a broad class of problems (including static risks) under the state-augmentation framework.

Apart from value-based approaches, there are also policy gradient methods for learning risk-sensitive policies for CVaR (Tamar et al., 2015b; Tang et al., 2019; Greenberg et al., 2022; Kim & Min, 2024; Schneider et al., 2024; Mead et al., 2025). These methods come with their own set of challenges like sample inefficiency and "blindness to success" (Greenberg et al., 2022; Mead et al., 2025). Moreover, the majority of these methods restrict their search for optimal policies, to the class of Markov policies, which is shown to be insufficient for optimizing static CVaR problems (Shapiro, 2009; Wang et al., 2025; Muni et al., 2025).

# 3. Background

**Notation.** Given a real number $x \in \mathbb{R}$, we denote the positive and negative part of $x$ by $x_+ := \max(x, 0)$ and $x_- := -\min(x, 0)$ respectively.

## 3.1. Risk-Neutral Reinforcement Learning

A discounted Markov decision process (MDP) $\mathcal{M}$ is a tuple $(\mathcal{S}, \mathcal{A}, P, r, \bar{s}_0, \gamma)$ such that $\mathcal{S}$ is a state-space, $\mathcal{A}$ an action space. The component $P : \mathcal{S} \times \mathcal{A} \to \Delta_{\mathcal{S}}$ models a transition probability $P_{s,a}$ over $\mathcal{S}$ from any state-action pair $(s, a) \in \mathcal{S} \times \mathcal{A}$, $r : \mathcal{S} \times \mathcal{A} \to [-r_{\max}, r_{\max}]$ is a bounded reward function, $\bar{s}_0$ is the initial state, and $\gamma \in [0, 1)$ a discount factor. The stochasticity of the process relies on state transitions and action decisions. Thus, a realized trajectory up to time $t$ is denoted by $\tau_{:t} := (s_0, a_0, \cdots, s_{t-1}, a_{t-1}, s_t)$, which belongs to the set of length-$t$ histories $\mathcal{T}_t$. Infinite-horizon trajectories are denoted by $\tau := (s_0, a_0, s_1, \cdots) \in \mathcal{T}$. A policy governs action decisions. Formally, it is a sequence of decision rules $\boldsymbol{\pi} := (\pi_t)_{t \in \mathbb{N}_0}$ with $\pi_t : \mathcal{T}_t \to \Delta_{\mathcal{A}}$. In that

case, the policy depends on the history and we write $\boldsymbol{\pi} \in \Pi^{\mathrm{H}}$. If it only depends on the current state, i.e., $\pi_t : \mathcal{S} \to \Delta_{\mathcal{A}}$, it is a Markov policy and we write $\boldsymbol{\pi} \in \Pi^{\mathrm{M}} \subset \Pi^{\mathrm{H}}$. In the remainder, we will denote by $\mathbb{P}_{\bar{s}_0}^{\boldsymbol{\pi}}(\tau)$ the distribution of a trajectory following any policy $\boldsymbol{\pi} \in \Pi^{\mathrm{H}}$ starting from $s_0 = \bar{s}_0$. The *risk-neutral* objective to maximize is the expected return $\mathbb{E}^{\boldsymbol{\pi}}[R(\tau)]$, where $R(\tau) := \sum_{t=0}^{\infty} \gamma^t r(s_t, a_t)$. In this case, optimality can be attained by a stationary policy $\boldsymbol{\pi} = (\pi_0, \pi_0, \cdots)$ (Puterman, 1994).

## 3.2. Conditional Value-at-Risk

For a bounded random variable $X$ of cumulative distribution $F_X(x) := \mathbb{P}[X \leq x]$, the conditional value-at-risk (CVaR) at confidence level $\alpha \in [0, 1]$ measures the expectation of $X$ on the lower tail distribution:

$$\mathrm{CVaR}_\alpha[X] := \mathbb{E}[X | X \leq \mathrm{VaR}_\alpha(X)],$$

where $\mathrm{VaR}_\alpha(X) = \max\{x : F_X(x) \leq \alpha\}$ (Hau et al., 2025). The CVaR conveniently admits a convex formulation (Rockafellar et al., 2000):[1]

$$\mathrm{CVaR}_\alpha[X] = \sup_{\eta \in \mathbb{R}} \left\{ \eta - \frac{1}{\alpha} \mathbb{E}[(\eta - X)_+] \right\}. \quad (1)$$

In this paper, we ultimately want to maximize the CVaR of the return, $\mathrm{CVaR}_\alpha^{\boldsymbol{\pi}}[R(\tau)]$, over the set of policies. Here, the superscript $\boldsymbol{\pi}$ implicitly means that CVaR is according to distribution $\tau \sim \mathbb{P}_{\bar{s}_0}^{\boldsymbol{\pi}}(\tau)$.

As we present in the following proposition, this CVaR maximization problem also admits an equivalent reformulation. The proof is in Appendix B.2.

**Proposition 3.1.** *The policy optimization of the CVaR MDP can be reformulated as:*

$$\max_{\boldsymbol{\pi} \in \Pi^H} CVaR_\alpha^{\bar{s}_0, \boldsymbol{\pi}}[R(\tau)]$$

$$= \sup_{z \in \mathbb{R}} \left\{ -z + \frac{1}{\alpha} \max_{\boldsymbol{\pi} \in \Pi^H} \mathbb{E}_{\tau \sim \mathbb{P}_{\bar{s}_0}^{\boldsymbol{\pi}}}[-(R(\tau) + z)_-] \right\} \quad (2)$$

$$= \sup_{z \in \mathbb{R}} \left\{ -z - \frac{z_-}{\alpha} + \right.$$

$$\left. \frac{1}{\alpha} \max_{\boldsymbol{\pi} \in \Pi^H} \left\{ \mathbb{E}_{\tau \sim \mathbb{P}_{\bar{s}_0}^{\boldsymbol{\pi}}}[-(R(\tau) + z)_-] + z_- \right\} \right\}. \quad (3)$$

Problems (2) and (3) show that the static CVaR maximization problem can be decomposed into an inner optimization over policies and an outer optimization over $\mathbb{R}$. The optimization over scalar parameter $z \in \mathbb{R}$ can be easily tackled once we find an optimal policy. However, as opposed to the risk-neutral objective, optimal policies in this setting are generally history-dependent (Bäuerle & Ott, 2011).

---

[1]See Appendix B.1 for detailed adaptation of this representation.

### 3.3. State Augmentation for CVaR

To solve the inner optimization in (2), Bäuerle & Ott (2011) proposed an elegant solution: augment the original state space with an auxiliary variable that tracks the cumulative reward over the trajectory. Specifically, the augmented state space $\bar{\mathcal{S}} := \mathcal{S} \times \mathbb{R}$, consists of the nominal state $s$ and a numerical threshold $z$ which tracks the accumulated reward so far along a trajectory. This *augmented* state encodes all necessary information of the history, and evolves deterministically according to

$$\bar{P}((s', z')|(s, z), a) := P(s'|s, a)\mathbf{1}\left(z' = \frac{r(s, a) + z}{\gamma}\right),$$

for all $(s, z), (s', z') \in \bar{\mathcal{S}}$. This augmented state $(s, z)$ is Markov and consequently allows for applying DP solutions to search over stationary policies that are provably optimal. Stationary policies on the augmented space are denoted by $\tilde{\boldsymbol{\pi}} := (\tilde{\pi}, \tilde{\pi}, \cdots) \in \tilde{\Pi}$. Bäuerle & Ott (2011) solved the inner problem in (2) using the augmented MDP via an optimal Bellman operator defined as:

$$[Tv](s, z) = \max_{a \in \mathcal{A}} \gamma \mathbb{E}_{s' \sim P_{s,a}}\left[v\left(s', \frac{r(s, a) + z}{\gamma}\right)\right],$$

for $(s, z) \in \bar{\mathcal{S}}$, with unique fixed point $v^*(s, z) := \mathbb{E}_{\tau \sim \mathbb{P}_s^\pi}[-(R(\tau) + z)_-]$.[2] The operator $T$ possesses desirable properties, namely satisfying a $\gamma$ contraction under sup-norm, whose fixed point characterizes an optimal policy $\tilde{\pi}$. However, these properties are only guaranteed in a very restricted set of unbounded functions that are nondecreasing in $z$, 1-Lipschitz, compactly supported in $z$, and can be expressed as a function $c(s) + z$ when $z < 0$.

When viewed on the more practical space of bounded functions, $T$ unfortunately has a non-informative fixed point, as formalized in the following lemma (proof in Appendix B.3).

**Lemma 3.2.** *The Bellman operator $T$ has the zero function $\mathbf{0}$ as its unique fixed point, in the set $\mathcal{L}_\infty(\bar{\mathcal{S}})$ of bounded functions.*

The above operator also inherently captures sparse rewards in the augmented MDP, raising challenges of temporal credit assignment (Pignatelli et al., 2024). These restrictive function class requirements and induced sparsity pose practical challenges, often necessitating heuristic reward capping (Mead et al., 2025) or customized risk curriculum for training of CVaR objectives (Greenberg et al., 2022). To address these limitations, we propose a transformed augmented MDP and Bellman operator that builds on the alternative reformulation (3) instead of (2).

---

[2]Bäuerle & Ott (2011) minimized the CVaR of a cost, so we adapted their formulation to our reward setting.

## 4. A New Augmented MDP for Static CVaR

In this section, we aim to solve $\max_{\boldsymbol{\pi} \in \Pi^H} \text{CVaR}_\alpha^{\bar{s}_0, \boldsymbol{\pi}}[R(\tau)]$ using the alternate bilevel formulation in (3). This yields a Bellman operator that (i) is *contracting on the full space of bounded functions*; and (ii) propagates an *informative per-step reward signal*. These properties will in turn enable us to devise risk-sensitive analogs of approximate DP algorithms including value iteration and a model-free Q-learning algorithm, based on discretization of the augmented state $z$.

Define the following functions for any $(s, z) \in \bar{\mathcal{S}}$:

$$\bar{v}^*(s, z) := \max_{\boldsymbol{\pi} \in \Pi^H} \mathbb{E}_{\tau \sim \mathbb{P}_s^\pi}[-(R(\tau) + z)_- + z_-].$$

A first important property is that $\bar{v}^*$ is bounded (proof in Appendix B.4), so it is well defined for the operator we introduce next. Moreover, $\bar{v}^*$ explicitly relates to the optimal CVaR, as formalized below.

**Proposition 4.1.** *For all $(s, z) \in \bar{\mathcal{S}}$, it holds that $|\bar{v}^*(s, z)| \leq \frac{r_{\max}}{1-\gamma}$. Furthermore, we have*

$$\max_{\boldsymbol{\pi} \in \Pi^H} CVaR_\alpha^{\bar{s}_0, \boldsymbol{\pi}}[R(\tau)] = \sup_{z \in \mathbb{R}}\left\{\frac{1}{\alpha}\bar{v}^*(\bar{s}_0, z) - z - \frac{z_-}{\alpha}\right\}.$$

This result is in sharp contrast with properties of the value function defined in (Bäuerle & Ott, 2011), which takes the form $v^*(s, z) := \max_{\bar{\boldsymbol{\pi}} \in \Pi^H} \mathbb{E}_{\tau \sim \mathbb{P}_s^\pi}[-(R(\tau) + z)_-] = \bar{v}^*(s, z) - z_-$ in our notation. Namely, $|v^*(s, z)| \to \infty$ as $z \to -\infty$, which requires a search in a space where all functions are unbounded.

Working in the space of bounded value functions is convenient for analysis purposes (Bertsekas, 1995). There, we can define the *CVaR-Bellman operator* $\bar{T} : \mathcal{L}_\infty(\bar{\mathcal{S}}) \to \mathcal{L}_\infty(\bar{\mathcal{S}})$:

$$[\bar{T}\bar{v}](s, z) := \max_{a \in \mathcal{A}}\Bigg\{\tilde{r}(s, z, a)$$
$$+ \gamma \mathbb{E}_{s' \sim P_{s,a}}\left[\bar{v}\left(s', \frac{r(s, a) + z}{\gamma}\right)\right]\Bigg\}, \quad (4)$$

with $\tilde{r}(s, z, a) := z_- - (r(s, a) + z)_-, \forall (s, z) \in \bar{\mathcal{S}}, a \in \mathcal{A}$.

Note that $\bar{T}$ captures the traditional Bellman operator in the augmented CVaR MDP proposed by Bäuerle & Ott (2011) but where the original augmented reward of zero is replaced with the more informative reward $\tilde{r}(s, z, a)$. The following theorem establishes contraction and fixed point properties of $\bar{T}$ on the space of bounded functions (proof in Appendix B.5).

**Theorem 4.2.** *The operator $\bar{T}$ is a $\gamma$-contraction with respect to the sup-norm, and admits the CVaR-function $\bar{v}^*$ as a unique fixed point in $\mathcal{L}_\infty(\bar{\mathcal{S}})$. Moreover, for all $(s, z) \in \bar{\mathcal{S}}$, it holds that $\bar{v}^*(s, z) = \bar{v}^*(s, \text{p}(z))$, where $\text{p}(x) := \max(-r_\gamma, \min(x, r_\gamma))$, and $r_\gamma := \frac{r_{\max}}{1-\gamma}$.*

Denoting by $\bar{v}_{\mathrm{p}}^*(s,z) := \bar{v}^*(s,\mathrm{p}(z))$ the projected version of optimal CVaR, we can deduce from Theorem 4.2 that $\bar{T}\bar{v}^*(s,z) = \bar{v}^*(s,z) = \bar{v}_{\mathrm{p}}^*(s,z) = \bar{T}\bar{v}_{\mathrm{p}}^*(s,z)$. As a result, we can indifferently work with CVaR-functions or their projected equivalent, effectively constraining $z$ to a bounded interval without impairing optimality.

Even when the original state space is tabular, the augmented value function remains challenging to handle because its second argument, $z \in [-r_\gamma, r_\gamma]$, is continuous. In the following result we show that $\bar{v}^*$ can be approximated accurately using a uniform discrete grid over $z$. This is made possible by two key properties: the superadditivity of $(x + y)_-$ and the fact that $\bar{v}^*$ is Lipschitz continuous in $z$.

# 5. Discretized Static CVaR Bellman Operators

We have established that one can project the continuous argument $z$ of the augmented CVaR on a bounded interval and still use Bellman operators to achieve optimality. We are yet to address the challenge of handling continuous augmented states. In the remainder of this paper, we solve static CVaR using finite discretization. This will allow us to design value iteration and Q-learning style algorithms in the following sections. First, we lay the foundations and assumptions that ensure the stability of the discretization scheme under Bellman updates. The granularity of the discretization naturally determines how accurately it approximates the true optimum. We then establish performance bounds between the optimal CVaR and its tabular approximation.

**Assumption 5.1.** The reward is non-positive, i.e., $r \leq 0$.

Following Assumption 5.1, we focus our analysis and experimentation on non-positive rewards. However, these results can also be adapted to non-negative rewards.[3] [4]

Consider two non-decreasing functions $\mathrm{l}, \mathrm{u} : \mathbb{R} \to \mathbb{R}$ bounding the identity, i.e., for all $x \in \mathbb{R}, \mathrm{l}(x) \leq x \leq \mathrm{u}(x)$. Using these two envelope functions, one can define respective bounding operators as:

$$\bar{T}^{\mathrm{e}}\bar{v}(s,z) := \max_{a \in \mathcal{A}} \bar{q}_{\mathrm{poe}}(s,z,a) \tag{5}$$

$$:= \max_{a \in \mathcal{A}} \left\{ \tilde{r}(s,z,a) + \gamma \mathbb{E}_{s' \sim P_{s,a}} \left[ \bar{v}_{\mathrm{poe}} \left( s', \frac{r(s,a)+z}{\gamma} \right) \right] \right\},$$

for all $(s,z) \in \bar{\mathcal{S}}$, where $\mathrm{e} \in \{\mathrm{l,p,u}\}$, and $\bar{v}_{\mathrm{poe}}(s,z) := \bar{v}(s,\mathrm{p}(\mathrm{e}(z)))$ and $\circ$ denotes function composition. One can further see that $\bar{T}^{\mathrm{e}}\bar{v}(s,z) = [\bar{T}\bar{v}_{\mathrm{poe}}](s,z)$.

---

[3]Specifically, Propositions 5.2 to 5.4 extend to the non-negative rewards using $\bar{T}^{\mathrm{e}}\bar{v}(s,z) := \max_{a \in \mathcal{A}} \bar{q}_{\mathrm{poe}^c}(s,z,a)$, where $\mathrm{l}^c := \mathrm{u} = [\mathrm{l}^c]^c$, with value functions $\bar{v}^{*,\mathrm{l}}$ and $\bar{v}^{*,\mathrm{u}}$ being non-increasing.

[4]Due to the translation invariance property of CVaR, one could replace $r(s,a)$ in given MDP with $r(s,a) - r_\gamma$, to obtain a new MDP with non-positive reward while preserving the same optimal CVaR policy.

When rewards are all non-positive, $\bar{T}^{\mathrm{l}}$ and $\bar{T}^{\mathrm{u}}$ have the useful property of preserving their respective form of ordering for non-decreasing value functions (proof in Appendix B.6).

**Proposition 5.2.** *Suppose Assumption 5.1 holds and let $\bar{v} \in \mathcal{L}_\infty(\bar{\mathcal{S}})$ be a non-decreasing function. Then, for all $k \geq 0$, it holds that $[\bar{T}^l]^k \bar{v} \leq [\bar{T}^{\mathrm{p}}]^k \bar{v} \leq [\bar{T}^u]^k \bar{v}$.*

Moreover, the bounding operators are contracting on the sup-norm and induce fixed points, as formalized in the following theorem (proof in Appendix B.7). These properties serve as the basis for designing tractable approximate value iteration that provides bounds for the true optimal value function.

**Theorem 5.3.** *Both operators $\bar{T}^l$ and $\bar{T}^u$ are $\gamma$-contracting for the sup-norm and admit unique fixed points $\bar{v}^{*,l}$ and $\bar{v}^{*,u}$, respectively. Moreover, if Assumption 5.1 is satisfied, then both value functions are non-decreasing in $z$ and:*

$$\bar{v}^* - \bar{\Delta}_l \leq \bar{v}^{*,l} \leq \bar{v}^* \leq \bar{v}^{*,u} \leq \bar{v}^* + \bar{\Delta}_u,$$

*where $\bar{\Delta}_e := \gamma(1-\gamma)^{-1} \sup_z |e(z) - z|$ for $e \in \{l, u\}$.*

Equipped with some $\bar{v}^{*,\mathrm{e}}$ and its associated $\bar{q}_{\mathrm{poe}}^{*,\mathrm{e}}(s,z,a)$ (see Eq. (5)), we can now derive an approximately optimal policy for the CVaR MDP as follows:

$$\bar{\pi}^{*,\mathrm{e}}(\tau_{:t}, z) := \tilde{\pi}^*(s, \zeta(\tau_{:t}, z)),$$

for all $\tau_{:t} \in \mathcal{T}_t, z \in \mathbb{R}$, where $\zeta(s, z) := z$, and for $t \geq 1$

$$\zeta(\tau_{:t}, z) := \mathrm{p} \circ \mathrm{e} \left( \frac{r(s_{t-1}, a_{t-1}) + \zeta(\tau_{:t-1}, z)}{\gamma} \right)$$

while

$$\tilde{\pi}^{*,\mathrm{e}}(s, z) := \operatorname*{argmax}_{a \in \mathcal{A}} \bar{q}_{\mathrm{poe}}^{*,\mathrm{e}}(s, z, a).$$

Recall that we are left with an outer optimization over $z$ to effectively deduce CVaR (3). Combining this optimization with lower and upper bounds yields the following guarantees (proof in Appendix B.8).

**Theorem 5.4.** *Let $\Psi^* := \max_{\pi \in \Pi^H} CVaR_\alpha^{\bar{s}_0, \pi}[R(\tau)]$, and*

$$\Psi^e := \sup_{z \in \mathbb{R}} \left\{ \frac{1}{\alpha} \left( \bar{v}^{*,e}(\bar{s}_0, z) - z_- \right) - z \right\}, \tag{6}$$

*with $e \in \{l, u\}$. If Assumption 5.1 is satisfied, then:*

$$\Psi^* - \bar{\Delta}_l/\alpha \leq \Psi^l \leq CVaR_\alpha^{\bar{s}_0, \bar{\pi}(\cdot, z_l^*)}[R(\tau)] \leq \Psi^*,$$

*and $\Psi^* \leq \Psi^u \leq \Psi^* + \bar{\Delta}_u/\alpha$, for any $z_l^*$ such that $\Psi^l = \left\{ \frac{1}{\alpha} \left( \bar{v}^{*,l}(\bar{s}_0, z_l^*) - (z_l^*)_- \right) - z_l^* \right\}$.*

We finally discuss a natural discretization based on a uniform grid of granularity $\Delta := r_\gamma/K$ for some $K \geq 1$. In that case, $\mathrm{l}_\Delta(x) := \max\{k\Delta : k \in \mathbb{Z}, k\Delta \leq x\}$ and $\mathrm{u}_\Delta(x) = \min\{k\Delta : k \in \mathbb{Z}, k\Delta \geq x\}$. By construction

$l_\Delta(x) \le x \le u_\Delta(x)$ for all $x \in \mathbb{R}$, both $l_\Delta$ and $u_\Delta$ are non-decreasing, and $\sup_z |l_\Delta(z) - z| = \sup_z |u_\Delta(z) - z| = \Delta$. This implies that Theorem 5.4 holds with $\{l_\Delta, u_\Delta\}$ and $\Delta_l = \Delta_u = \gamma(1-\gamma)^{-1}\Delta$. Furthermore, one can show that both $\bar{T}^{l_\Delta}$ and $\bar{T}^{u_\Delta}$ can be applied in the finite augmented space $\mathcal{S} \times \mathcal{Z}_\Delta$ with $\mathcal{Z}_\Delta := \{\Delta k : k \in \mathbb{Z}\} \cap [-r_\gamma, r_\gamma] \supseteq \{-r_\gamma, r_\gamma\}$ instead of $\bar{\mathcal{S}} = \mathcal{S} \times \mathbb{R}$ (proof in Appendix B.9).

**Proposition 5.5.** *For $e = \{l_\Delta, u_\Delta\}$, let $\hat{v}^{*,e}$ be the unique fixed point of $\bar{T}^e$ in $\mathcal{L}_\infty(\mathcal{S} \times \mathcal{Z}_\Delta)$, then $\bar{v}^{*,e}(s,z) = \bar{T}^e \hat{v}^{*,e}(s,z)$ for all $(s,z) \in \bar{\mathcal{S}}$.*

# 6. Approximate Static CVaR Algorithms

Theorem 5.4 and Proposition 5.5 provide the means to approximate, from above or below, the optimal solution of the CVaR MDP using an augmented tabular MDP with risk neutral objective. Such MDP takes the form $\tilde{\mathcal{M}} := (\mathcal{S} \times \mathcal{Z}_\Delta, \mathcal{A}, \tilde{P}, \tilde{r}, (\bar{s}_0, \nu), \gamma)$ with

$$\tilde{P}((s',z')|(s,z),a) :=$$
$$P(s'|s,a)\mathbf{1}\left(z' = p\left(e\left(\frac{r(s,a)+z}{\gamma}\right)\right)\right),$$

where $e \in \{l_\Delta, u_\Delta\}$ depending on whether a lower or upper bounding approximation, respectively, is needed. This augmented MDP formulation motivates the use of Q-value iteration and Q-learning algorithms depending on whether one knows the underlying MDP model $\mathcal{M}$ or can only interact with this environment. In this section, we propose a variant of each of these algorithms in order to identify the Q-value function $\hat{q}^{*,e}$ defined for all $(s,z) \in \mathcal{S} \times \mathcal{Z}_\Delta$ as:

$$\hat{q}^{*,e}(s,z,a) := \tilde{r}(s,z,a)$$
$$+ \gamma \mathbb{E}_{s' \sim P_{s,a}}\left[\hat{v}^{*,e}\left(s', p\left(e\left(\frac{r(s,a)+z}{\gamma}\right)\right)\right)\right],$$

which can in turn be used to produce $\hat{v}^{*,e} := \max_{a \in \mathcal{A}} \hat{q}^{*,e}(s,z,a)$. In a model-based setting, this gives access to $\bar{v}^{*,e} = \bar{T}^e \hat{v}^{*,e}$, $z_e^*$, $\bar{\pi}^{*,e}(\cdot,)$, and $\Psi^e$ with the guarantees offered by Theorem 5.4. Alternatively, in a model-free setting, one can derive approximations of $\bar{v}^{*,e}(s,z) \approx \hat{v}^{*,e}(s,p(e(z)))$, $z_e^* \approx \operatorname{argmax}_{z \in \mathcal{Z}_\Delta} \left\{\frac{1}{\alpha}(\hat{v}^{*,e}(\bar{s}_0,z) - (z)_-) - z\right\}$ and associated approximate $\bar{\pi}^{*,e}$ and $\Psi^e$.

## 6.1. Static Risk-Sensitive Value Iteration

We present in Algorithm 1 an implementation of Q-value iteration for the risk-neutral augmented MDP $\tilde{\mathcal{M}}$. The algorithm simultaneously estimates the globally optimal Q-function $\bar{q}^{*,e}$ for all initial states $s \in \mathcal{S}$ and all possible budget levels $z \in \mathcal{Z}_\Delta$. (Proof in Appendix B.10).

**Proposition 6.1.** *The function $\hat{q}_k^e$ returned by Algorithm 1 satisfies: $\|\hat{q}^{*,e} - \hat{q}_k^e\|_\infty \le \gamma(1-\gamma)^{-1}\epsilon$.*

---

**Algorithm 1** Static CVaR Value Iteration

**Require:** Grid $\mathcal{Z}_\Delta$, projection and rounding maps p and $e \in \{l_\Delta, u_\Delta\}$, discount $\gamma \in (0,1)$, convergence tolerance $\delta > 0$.
1: **Initialize:** $\hat{q}_0^e(s,z,a) = 0$ for all $(s,z,a)$
2: Set iteration counter $k \leftarrow 0$
3: **repeat**
4:     $\delta \leftarrow 0$
5:     **for** each $s \in \mathcal{S}$, $a \in \mathcal{A}$, $z \in \mathcal{Z}_\Delta$ **do**
6:         $z' \leftarrow p \circ e(\gamma^{-1}(r(s,a)+z))$
7:         $\tilde{r} \leftarrow z_- - (r(s,a)+z)_-$
        $\hat{q}_{k+1}^e(s,z,a) \leftarrow$
8: $$\tilde{r} + \gamma \sum_{s' \in \mathcal{S}} P(s'|s,a) \max_{a' \in \mathcal{A}} \hat{q}_k^e(s',z',a')$$
9:         $\delta \leftarrow \max\{\delta, |\hat{q}_{k+1}^e(s,z,a) - \hat{q}_k^e(s,z,a)|\}$
10:     **end for**
11:     $k \leftarrow k+1$
12: **until** $\delta < \epsilon$
13: **Return:** $\hat{q}_k^e$

---

Once $\hat{q}^{*,e}$ (and implicitly $\bar{v}^{*,e}$ and $\bar{q}^{*,e}$) is obtained, the outer optimization over $z$ (see Eq. (6) and Algorithm 3) can be done using global optimization on $z$. The policy execution pseudo-code is shown in Algorithm 4 in Appendix C.

## 6.2. Static Risk-Sensitive Q-Learning

We now propose a model-free Q-learning algorithm (Watkins & Dayan, 1992) for static CVaR that learns the optimal action-value function from samples. The pseudo-code is shown in Algorithm 2. A distinctive feature of this approach is that we employ a block-update for $z$ for improved data-efficiency. In standard Q-learning approaches (Watkins & Dayan, 1992; Tsitsiklis, 1994), it is common to draw one transition sample from the environment and update its Q-value using a TD-update. In the augmented-state setting, given a $(s,z,a,r)$ sample, we can deterministically compute the next $z'$ reached from any $z \in \mathcal{Z}_\Delta$ using $p(e(\gamma^{-1}(r+z)))$. Notice however that the "trajectory" of $z$ under any policy is determined by how $z_0$ is initialized (here from a uniform distribution). Since Q-learning is off-policy, once a single episode from the environment is collected, we can leverage the idea of data-relabeling (Andrychowicz et al., 2017; Eysenbach et al., 2020) to generate new samples for the training, without interacting with the environment. The key idea is that we can replay the episode under the assumption that it started from a different initial $z$ value. This leads to the block update scheme as shown in Algorithm 2 for which Theorem 6.2 provides sufficient conditions under which it converges to $\hat{q}^{*,e}$ (proof in Appendix B.11).

**Theorem 6.2.** *Assume the following conditions hold:*

*(1) (Robbins–Monro Step-sizes). The step-size schedule*

**Algorithm 2** Static CVaR Q-Learning

**Require:** Grid $\mathcal{Z}_\Delta$, projection and rounding maps p and $e \in \{l_\Delta, u_\Delta\}$, discount $\gamma \in (0, 1)$, initial state distribution $\nu$, episodes $M$, exploration schedule $\{\varepsilon_k\}$, step-size schedule via visitation counts $\beta(n)$.
1: **Initialize:** $\hat{q}_0^e(s, z, a) = 0$ for all $(s, z, a)$ and $N_0(s, a) = 0$ for all $(s, a)$
2: Set global-step counter $k \leftarrow 0$
3: **for** $i \leftarrow 1$ to $M$ **do**
4:     Sample initial state $s_k \sim \nu$
5:     Initial budget $z_k \sim \text{Unif}(\mathcal{Z}_\Delta)$
6:     **while** $s$ is not terminal **do**
7:         Choose $a_k$ $\varepsilon_k$-greedy w.r.t. $\{\hat{q}_k^e(s_k, z_k, a)\}_{a \in \mathcal{A}}$
8:         Execute $a_k$, observe $(r_k, s'_k)$ with $s'_k \sim P_{s_k, a_k}$
9:         $\hat{q}_{k+1}^e \leftarrow \hat{q}_k^e, \quad N_{k+1} \leftarrow N_k$
10:        $N_{k+1}(s, a) \leftarrow N_k(s, a) + 1$
11:        **for** $\tilde{z} \in \mathcal{Z}_\Delta$ **do**
12:           $\tilde{z}' \leftarrow \text{p} \circ \text{e}\big(\gamma^{-1}(r_k + \tilde{z})\big)$
13:           $\tilde{r} \leftarrow \tilde{z}_- - (r_k + \tilde{z})_-$
14:           $\hat{q}_{k+1}^e(s_k, \tilde{z}, a_k) \leftarrow \hat{q}_k^e(s_k, \tilde{z}, a_k) + \beta(N_k(s_k, a_k))\Big(\tilde{r} + \gamma \max_{a'} \hat{q}_k^e(s'_k, \tilde{z}', a') - \hat{q}_k^e(s_k, \tilde{z}, a_k)\Big)$
15:        **end for**
16:        Update budget: $z_{k+1} \leftarrow \text{p} \circ \text{e}\big(\gamma^{-1}(r_k + z_k)\big)$
17:        $s_{k+1} \leftarrow s'_k, \quad k \leftarrow k + 1$
18:     **end while**
19: **end for**
20: **Return** $\hat{q}_k^e$

$\beta(n)$ *satisfies:*
$$\sum_{n=0}^{\infty} \beta(n) = \infty, \qquad \sum_{n=0}^{\infty} \beta(n)^2 < \infty.$$

*(2) (**Sufficient Exploration**). Every nominal pair $(s, a) \in \mathcal{S} \times \mathcal{A}$ is visited infinitely often by the $\varepsilon$-greedy policy of Algorithm 2, i.e.,*
$$\lim_{k \to \infty} N_k(s, a) = \infty \text{ almost surely, } \forall (s, a) \in \mathcal{S} \times \mathcal{A}.$$

*Then, with probability one, the iterates $\{\hat{q}_k^e\}_{k=0}^{\infty}$ produced by Algorithm 2 converge in sup-norm to $\hat{q}^{*,e}$ as $k \to \infty$.*

Once the optimal action-value function $\hat{q}^{*,e}$ is learned, an approximate outer optimization (Algorithm 3) can be applied to obtain $\hat{z}^{*,e}$, which together define an approximately optimal CVaR policy through the policy execution described in Algorithm 4.

## 7. Empirical Results

We validate the proposed static CVaR algorithms on a stochastic gridworld, where a robot has to traverse a 2D ter-

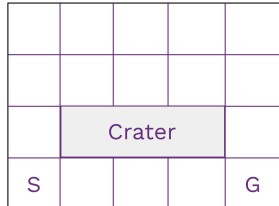

*Figure 2.* Stochastic Gridworld Environment.

rain map (Figure 2) starting at a safe position "S" and safely reaching the goal position "G" while being fuel-efficient. The robot can move in four cardinal directions; due to noise in sensing and control, there is a non-zero chance that an action leads to a move along an unintended direction. Each step receives a penalty of $-1$ to indicate the fuel usage. Gray cells denote craters with uneven terrain that may impair the robot and yield a penalty of $-10$. The goal state is absorbing, i.e. once the robot enters state "G", it remains there in perpetuity with a 0 reward. This reward scheme fulfills the non-positive reward requirements of Assumption 5.1. The objective is to compute a *safe*, crater-free path to the goal while being *fuel efficient*.

Both, static CVaR Q-value iteration (Q-VI) and Q-learning algorithms compute the optimal action-value function for all discrete budget levels simultaneously, and therefore need to be run only once to estimate the optimal Q-function $\hat{q}^{*,e}$ (Section 6). Using a grid size of $|\mathcal{Z}_\Delta| = 5k$, we run both algorithms with 10 independent seeds and compute the corresponding optimal starting budget for a spectrum of $\alpha$ values from near-robust ($\alpha = 0.01$) to risk-neutral ($\alpha = 1.0$) (pseudo-code in Algorithm 3). For each risk-averse policy induced by an optimal starting budget, we collect 10k rollouts with Algorithm 4, and use them to compare performance of both algorithms for varying risk-levels (Figure 3).

Figure 3a compares the empirical return distributions. Notice that, for both methods the lower tail of the distribution, representing the unsafe (high penalty) trajectories, shrinks as $\alpha : 1 \to 0$, indicating an increasingly risk-averse behavior. Figure 3b compares the empirical $\text{CVaR}_\alpha$ performance under each method versus the $\text{CVaR}_\alpha$ performance under a risk-neutral policy at the same risk-level $\alpha$. For small $\alpha$, both risk-averse agents outperform the risk-neutral policy. Figure 3c reports the *discounted* total number of crater entries along each trajectory. Under static risks, performance depends on the full return, which implies that due to discounting, crater entries later in the episode will yield a lower penalty than earlier ones. We see fewer crater entries, as risk-aversion increases. Both our methods agree closely across most $\alpha$, except around $\alpha \in [0.5, 0.6]$ where Q-learning behavior deviates slightly and shows higher variance. This region coincides with a policy switch where the agent starts to prefer the longer route to the goal instead of the bottom

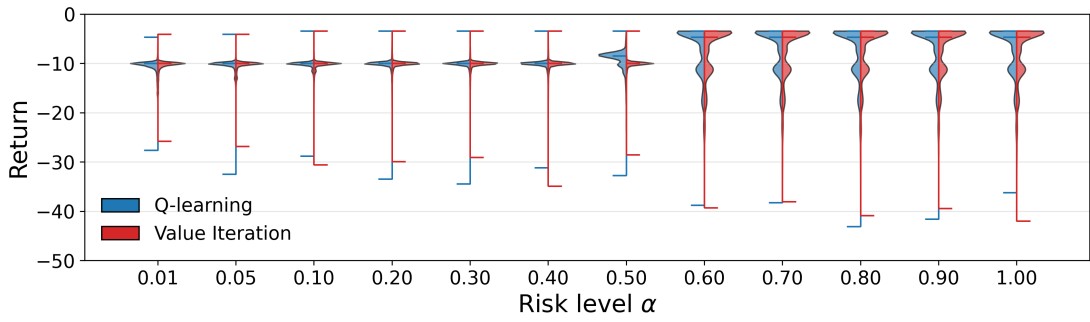

*(a)* Return distribution under optimal CVaR policies computed using risk-sensitive Q-learning and Q-VI algorithms.

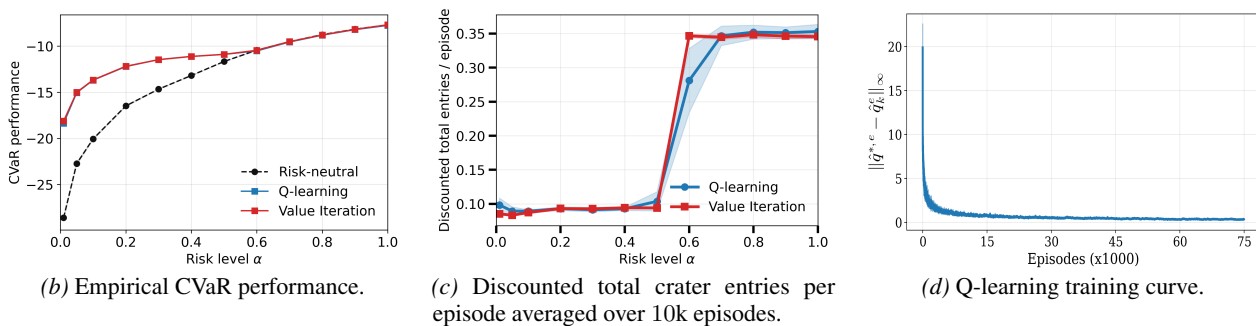

*(b)* Empirical CVaR performance.

*(c)* Discounted total crater entries per episode averaged over 10k episodes.

*(d)* Q-learning training curve.

*Figure 3.* Performance comparison between the proposed static CVaR Q-learning and Q-VI algorithms for various risk-levels $\alpha$.

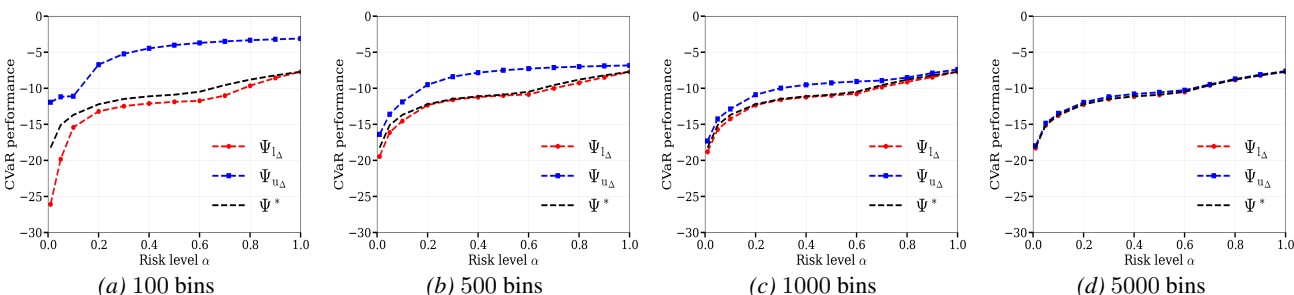

*(a)* 100 bins

*(b)* 500 bins

*(c)* 1000 bins

*(d)* 5000 bins

*Figure 4.* Comparing the effect of augmented state discretization resolution on the accuracy of CVaR performance. As resolution increases (left to right), the performance curves under the upper and lower bounding Bellman operators converge to the true value.

corridor. As risk aversion increases, the agent increasingly prefers a longer, safer route that reduces crater-entry risk. Figure 3d compares difference between the learned value function $\hat{q}_k^e$ at training iteration $k$ and the optimal Q-value estimated by static CVaR Q-VI. We see that after 50k training iterations, static CVaR Q-learning has converged.

Finally, to validate our theoretical results corresponding to the performance accuracy under the discretization, we run value iteration with the upper and lower Bellman operators $T^e$ and compare their performances, labeled $\psi_{u_\Delta}$ and $\psi_{l_\Delta}$ respectively, in Figure 4. The optimal performance value $\psi^*$, is computed by running Q-VI with 10k discrete $z$ values and plot the corresponding performance. Results in Figure 4 empirically validate the claims in Theorem 5.4; showing that the performance under the upper/lower bound operators maintain their order and performance gap between them

vanishes with finer discretization. (Experiment details are provided in Appendix D).

## 8. Conclusion and Future Directions

This paper revisits a foundational result in infinite horizon static CVaR MDPs (Bäuerle & Ott, 2011) and shows that several long-standing limitations like sparse rewards and restricted convergence guarantees, are not inherent to the static CVaR problem, but a consequence of the particular modeling choice. We propose an *exact* reformulation of the static CVaR objective, and derive a new Bellman operator that distributes CVaR-relevant information across time and eliminates the artificial sparsity introduced by prior formulations. Our operator enjoys global convergence guarantees without requiring carefully engineered initializations or admissibility constraints. Building on these theoretical results,

we develop two risk-averse approximate DP algorithms, analogous to standard VI and Q-learning. To operate on finite MDPs, we leverage a uniform discretization scheme and provide error bounds that quantify the impact of value-function approximation on the CVaR performance. We also prove the convergence of the static CVaR Q-learning algorithm under standard assumptions.

A limitation of our work is that we focus on tabular settings for our empirical analysis. We note however that the static CVaR Bellman operator $\bar{T}$ applies more generally and lays the groundwork for developing scalable risk-sensitive RL algorithms with function approximation. A key advantage of combining our algorithms with function approximators, such as deep neural networks, is that it will allow to directly represent the continuous augmented state with no restrictions on the sign of the reward functions.

## Impact Statement

This paper presents work whose goal is to advance the field of reinforcement learning. There are many potential societal consequences of our work, none of which we feel must be specifically highlighted here.

## Acknowledgments

The authors would like to thank Harley Wiltzer and Borna Sayedana for their feedback on an early draft of this paper. We would also like to thank Nima Akbarzadeh, Kaustubh Mani, Niki Howe, Ryan D'Orazio, Arnav Jain, Yudong Luo, Tianwei Ni and Glen Berseth for helpful discussions and insightful suggestions. Aneri Muni was partially funded by IVADO through its Machine Learning regroupement. Erick Delage was partially supported by the Canadian Natural Sciences and Engineering Research Council [Grant RGPIN-2022-05261] and by the Canada Research Chair program [950-230057].

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

## A. Illustration of Various Reward Schemes

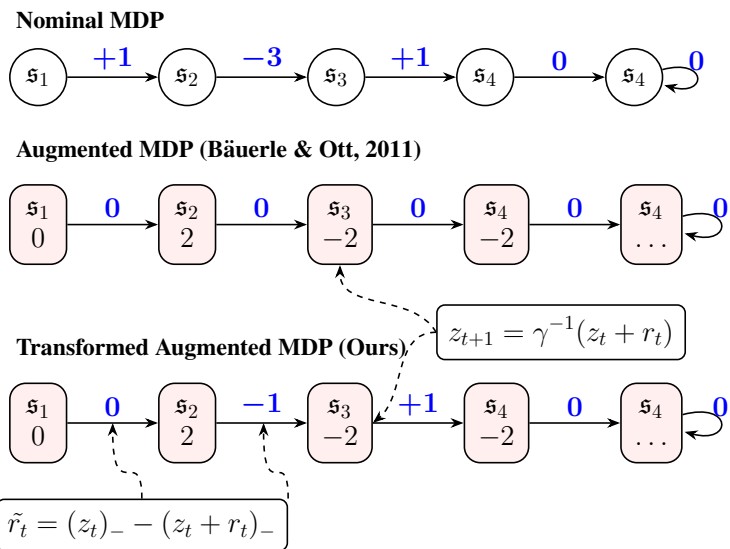

*Figure 5.* Illustration of the reward schemes produced along a trajectory under the augmented MDP of Bäuerle & Ott (2011) versus ours.

Figure 5 compares a trajectory under the nominal MDP, augmented MDP proposed by (Bäuerle & Ott, 2011), which we will refer to as B-AMDP and finally our transformed augmented MDP, which we call T-AMDP. (Figure 5 is a more detailed version of Figure 1). We consider a simple four state MDP where $\mathfrak{s}_4$ is an absorbing state, i.e. once the agent enters this state, it remains there in perpetuity, receiving a reward of 0. The numbers in blue represent the per-step rewards. The pink states represent the augmented states where the second component is the augmented dimension $z_t$, which keeps track of the cumulated reward scaled by $1/\gamma^t$ when initialized at $z_0 := 0$, along the trajectory.

As explained in Section 3.3, under B-AMDP, there are no per-stage rewards except for the end of the horizon $\hat{\mathcal{T}}$, where reward is given by $-(z_{\hat{\mathcal{T}}} + r_{\hat{\mathcal{T}}})_-$. In the infinite-horizon case, $\hat{\mathcal{T}} \to \infty$, thus the agent in B-AMDP effectively never gets the reward feedback.

B-AMDP's augmented state $z$ is updated according to the rule: $z_{t+1} = \gamma^{-1}(z_t + r_t)$. We use the same state-augmentation scheme as that of Bäuerle & Ott (2011), however, under our CVaR reformulation (Proposition 3.1), the per-step reward in the T-AMDP is given by $\tilde{r}_t := -(z_t + r_t)_- + (z_t)_-$.

Table 1 presents the calculations corresponding to the augmented state and reward values shown in Figure 5, we let initial state $\bar{s}_0 := \mathfrak{s}_1$, the initial augmented state $z_0 := 0$, and the discount factor $\gamma := 0.5$. We denote the reward under B-AMDP as $r^B$ and the rewards under T-AMDP as $\tilde{r}$.

*Table 1.* Details about the trajectory presented in Figure 5.

| t | $s_t$ | $r_t$ | $z_t$ | $z_{t+1}$ | $r_t^B$ | $\tilde{r}_t$ |
|---|---|---|---|---|---|---|
| 0 | $\mathfrak{s}_0$ | $+1$ | 0 | $(0.5)^{-1}(0+1) = 2$ | 0 | $\min(0, 0+1) - \min(0, 0) = 0$ |
| 1 | $\mathfrak{s}_1$ | $-3$ | 2 | $(0.5)^{-1}(2-3) = -2$ | 0 | $\min(0, 2-3) - \min(0, 2) = -1$ |
| 2 | $\mathfrak{s}_2$ | $+1$ | $-2$ | $(0.5)^{-1}(-2+1) = -2$ | 0 | $\min(0, -2+1) - \min(0, -2) = +1$ |
| 3 | $\mathfrak{s}_4$ | 0 | $-2$ | $(0.5)^{-1}(-2+0) = -4$ | 0 | $\min(0, -2+0) - \min(0, -2) = 0$ |
| ... | $\mathfrak{s}_4$ | 0 | $\gamma^{-(t-4)}(-4)$ | $\gamma^{-(t-3)}(-4)$ | 0 | 0 |

# B. Proofs of Theoretical Results

## B.1. Derivation of Equation (1) from (Rockafellar et al., 2000)

In (Rockafellar et al., 2000), the authors define a CVaR measure applied on losses $Y$ and that quantifies the amount of risk exposure as:

$$\overline{\text{CVaR}}_\beta[Y] := \inf_s s + \frac{1}{1-\beta}\mathbb{E}\big[(Y - s)^+\big],$$

where $\beta \in [0, 1]$ is the level of risk aversion. We rather employ a formulation where $X$ is a return, the measure quantifies risk protection, and $1 - \alpha$ captures the risk aversion. Hence,

$$\text{CVaR}_\alpha[X] := -\overline{\text{CVaR}}_{1-\alpha}[-X] = -\inf_s s + \frac{1}{\alpha}\mathbb{E}\big[(-X - s)^+\big] = \sup_s -s - \frac{1}{\alpha}\mathbb{E}\big[(-X - s)^+\big] = \sup_\eta \eta - \frac{1}{\alpha}\mathbb{E}\big[(\eta - X)^+\big].$$

## B.2. Proof of Proposition 3.1

*Proof.* We write the problem:

$$\max_{\boldsymbol{\pi}\in\Pi^{\text{H}}} \text{CVaR}_\alpha^{\bar{s}_0,\boldsymbol{\pi}}[R(\tau)] \text{ with } R(\tau) = \sum_{t=0}^\infty \gamma^t r(s_t, a_t).$$

Based on the CVaR convex form (1), we can write:

$$
\begin{aligned}
\max_{\boldsymbol{\pi}\in\Pi^{\text{H}}} \text{CVaR}_\alpha^{\bar{s}_0,\boldsymbol{\pi}}[R(\tau)] &= \max_{\boldsymbol{\pi}\in\Pi^{\text{H}}} \sup_{\eta\in\mathbb{R}} \left\{ \eta - \frac{1}{\alpha}\mathbb{E}_{\tau\sim\mathbb{P}_{\bar{s}_0}^{\boldsymbol{\pi}}}[(\eta - R(\tau))_+] \right\} \\
&= \max_{\boldsymbol{\pi}\in\Pi^{\text{H}}} \sup_{z\in\mathbb{R}} \left\{ -z - \frac{1}{\alpha}\mathbb{E}_{\tau\sim\mathbb{P}_{\bar{s}_0}^{\boldsymbol{\pi}}}[(-z - R(\tau))_+] \right\} && [\text{Set } z := -\eta] \\
&= \max_{\boldsymbol{\pi}\in\Pi^{\text{H}}} \sup_{z\in\mathbb{R}} \left\{ -z + \frac{1}{\alpha}\mathbb{E}_{\tau\sim\mathbb{P}_{\bar{s}_0}^{\boldsymbol{\pi}}}[\min(0, R(\tau) + z)] \right\} && [\text{Use } \max(a,b) = -\min(-a,-b)] \\
&= \sup_{z\in\mathbb{R}} \max_{\boldsymbol{\pi}\in\Pi^{\text{H}}} \left\{ -z + \frac{1}{\alpha}\mathbb{E}_{\tau\sim\mathbb{P}_{\bar{s}_0}^{\boldsymbol{\pi}}}[-(R(\tau) + z)_-] \right\} && [\min(a,0) = -a_-] \\
&= \sup_{z\in\mathbb{R}} \{ -z + \frac{1}{\alpha}\max_{\boldsymbol{\pi}\in\Pi^{\text{H}}} \mathbb{E}_{\tau\sim\mathbb{P}_{\bar{s}_0}^{\boldsymbol{\pi}}}[-(R(\tau) + z)_-] \} \\
&= \sup_{z\in\mathbb{R}} \left\{ -z + \frac{1}{\alpha}\left\{ \max_{\boldsymbol{\pi}\in\Pi^{\text{H}}} \mathbb{E}_{\tau\sim\mathbb{P}_{\bar{s}_0}^{\boldsymbol{\pi}}}[-(R(\tau) + z)_-] + z_- - z_- \right\} \right\} \\
&= \sup_{z\in\mathbb{R}} \left\{ -z + \frac{1}{\alpha}\left( -z_- + \max_{\boldsymbol{\pi}\in\Pi^{\text{H}}} \left\{ \mathbb{E}_{\tau\sim\mathbb{P}_{\bar{s}_0}^{\boldsymbol{\pi}}}[-(R(\tau) + z)_-] + z_- \right\} \right) \right\},
\end{aligned}
$$

so we established the two relations. $\qquad\square$

## B.3. Proof of Theorem 3.2

*Proof.* For completeness, we start by confirming the $\gamma$-contraction property of $T$. Namely, given any $v_1, v_2 : \bar{\mathcal{S}} \to \mathbb{R}$, we have that for all $(s, z) \in \bar{\mathcal{S}}$:

$$
\begin{aligned}
[Tv_1](s,z) - [Tv_2](s,z) &= \max_{a\in\mathcal{A}} \gamma\mathbb{E}_{s'\sim P_{s,a}}\left[ v_1\left(s', \frac{r(s,a) + z}{\gamma}\right) \right] - \max_{a\in\mathcal{A}} \gamma\mathbb{E}_{s'\sim P_{s,a}}\left[ v_2\left(s', \frac{r(s,a) + z}{\gamma}\right) \right] \\
&\leq \max_{a\in\mathcal{A}} \gamma\mathbb{E}_{s'\sim P_{s,a}}\left[ v_1\left(s', \frac{r(s,a) + z}{\gamma}\right) \right] - \gamma\mathbb{E}_{s'\sim P_{s,a}}\left[ v_2\left(s', \frac{r(s,a) + z}{\gamma}\right) \right] \\
&= \max_{a\in\mathcal{A}} \gamma\mathbb{E}_{s'\sim P_{s,a}}\left[ v_1\left(s', \frac{r(s,a) + z}{\gamma}\right) - v_2\left(s', \frac{r(s,a) + z}{\gamma}\right) \right] \\
&\leq \max_{a\in\mathcal{A}} \gamma\mathbb{E}_{s'\sim P_{s,a}}[\|v_1 - v_2\|_\infty] = \gamma\|v_1 - v_2\|_\infty.
\end{aligned}
$$

A similar argument confirms also that:

$$[Tv_2](s,z) - [Tv_1](s,z) \le \gamma \|v_1 - v_2\|_\infty, \ \forall (s,z) \in \bar{\mathcal{S}},$$

hence confirming the $\gamma$-contraction property under the sup-norm : $\|Tv_2 - Tv_1\|_\infty \le \gamma \|v_2 - v_1\|_\infty$.

One can also easily confirm that $\bar{v}_0(s,z) := 0$ is a fixed point of $T$:

$$[T\bar{v}_0](s,z) = \max_{a \in \mathcal{A}} \gamma \mathbb{E}_{s' \sim P_{s,a}} \left[ \bar{v}_0 \left( s', \frac{r(s,a)+z}{\gamma} \right) \right] = \max_{a \in \mathcal{A}} \gamma \mathbb{E}_{s' \sim P_{s,a}} [0] = \bar{v}_0(s,z), \ \forall (s,z) \in \bar{\mathcal{S}}.$$

To show that $0$ is the only fixed point in $\mathcal{L}_\infty$, let $\hat{v}$ be any fixed point in $\mathcal{L}_\infty$. Then, we must have that

$$\|\hat{v} - \bar{v}_0\|_\infty = \|T\hat{v} - T\bar{v}_0\|_\infty \le \gamma \|\hat{v} - \bar{v}_0\|_\infty \ \Rightarrow \ (1-\gamma)\|\hat{v} - \bar{v}_0\|_\infty \le 0 \ \Rightarrow \ \|\hat{v} - \bar{v}_0\|_\infty \le 0,$$

since $0 \le \gamma < 1$ and $\|\hat{v} - \bar{v}_0\|_\infty \le \|\hat{v}\|_\infty + \|\bar{v}_0\|_\infty < \infty$. $\qquad \square$

## B.4. Proof of Proposition 4.1

*Proof.* Let's first prove that $|\bar{v}^*(s,z)| \le \frac{r_{\max}}{1-\gamma}, \quad \forall (s,z) \in \bar{\mathcal{S}}$. For an infinite-horizon trajectory $\tau \in \mathcal{T}$, define:

$$f(\tau, z) := -(R(\tau)+z)_- + z_- = \min(R(\tau)+z, 0) - \min(z, 0) = \begin{cases} \min(R(\tau)+z, 0) \text{ if } z \ge 0, \\ \min(R(\tau), -z), \text{ otherwise.} \end{cases}$$

This implies that

$$\begin{aligned} f(\tau, z) &\le \sup_{z \in \mathbb{R}} f(\tau, z) \\ &= \max \left( \sup_{z \ge 0} \{\min(R(\tau)+z, 0)\}, \sup_{z < 0} \{\min(R(\tau), -z)\} \right) \\ &= \max(0, R(\tau)) \\ &\le |R(\tau)| \le \sum_{t=0}^\infty \gamma^t |r(s_t, a_t)| \le \frac{1}{1-\gamma} r_{\max}. \end{aligned}$$

On the other hand,

$$\begin{aligned} f(\tau, z) &\ge \inf_{z \in \mathbb{R}} f(\tau, z) \\ &= \min \left( \inf_{z \ge 0} \{\min(R(\tau)+z, 0)\}, \inf_{z < 0} \{\min(R(\tau), -z)\} \right) \\ &= \min \{\min(R(\tau), 0), \min(R(\tau), 0)\} \\ &\ge -|R(\tau)| \ge -\sum_{t=0}^\infty \gamma^t |r(s_t, a_t)| \ge -\frac{1}{1-\gamma} r_{\max}. \end{aligned}$$

We thus conclude that $|f(\tau, z)| \le \frac{1}{1-\gamma} r_{\max}$. Since $\bar{v}^*(s,z) = \max_{\boldsymbol{\pi} \in \Pi^{\text{H}}} \mathbb{E}_{\tau \sim \mathbb{P}_s^\pi} [-(R(\tau)+z)_- + z_-] = \max_{\boldsymbol{\pi} \in \Pi^{\text{H}}} \mathbb{E}_{\tau \sim \mathbb{P}_s^\pi} [f(\tau, z)]$, it results that:

$$|\bar{v}^*(s,z)| = |\max_{\boldsymbol{\pi} \in \Pi^{\text{H}}} \mathbb{E}_{\tau \sim \mathbb{P}_s^\pi}[f(\tau, z)]| \le \max_{\boldsymbol{\pi} \in \Pi^{\text{H}}} \mathbb{E}_{\tau \sim \mathbb{P}_s^\pi}[|f(\tau, z)|] \le \frac{1}{1-\gamma} r_{\max}.$$

We now prove the second statement in the proposition. Based on Eq. (3), the optimal CVaR is:

$$\begin{aligned} \max_{\boldsymbol{\pi} \in \Pi^{\text{H}}} \text{CVaR}_\alpha^{\bar{s}_0, \boldsymbol{\pi}}[R(\tau)] &= \sup_{z \in \mathbb{R}} \left\{ -z - \frac{z_-}{\alpha} + \frac{1}{\alpha} \max_{\boldsymbol{\pi} \in \Pi^{\text{H}}} \left\{ \mathbb{E}_{\tau \sim \mathbb{P}_{\bar{s}_0}^\pi}[-(R(\tau)+z)_-] + z_- \right\} \right\} \\ &= \sup_{z \in \mathbb{R}} \left\{ -z - \frac{z_-}{\alpha} + \frac{1}{\alpha} \bar{v}^*(\bar{s}_0, z) \right\}. \end{aligned}$$

$\qquad \square$

### B.5. Proof of Theorem 4.2

This proof will follow easily after establishing the contraction property of $T$, that $\bar{v}^*$ is a fixed point, and that it is constant outside the interval $[-r_\gamma, r_\gamma]$. This is divided into three lemmas.

**Lemma B.1.** *The operator $\bar{T}$ is a $\gamma$-contraction with respect to the sup-norm.*

*Proof.* For any functions $v_1, v_2 \in \mathcal{L}_\infty(\bar{\mathcal{S}})$, we have:

$$
\begin{aligned}
[\bar{T}v_1](s,z) - [\bar{T}v_2](s,z) &= \max_{a \in \mathcal{A}} \left\{ \tilde{r}(s,z,a) + \gamma \mathbb{E}_{s' \sim P_{s,a}} \left[ v_1 \left( s', \frac{r(s,a)+z}{\gamma} \right) \right] \right\} \\
&\quad - \max_{a \in \mathcal{A}} \left\{ \tilde{r}(s,z,a) + \gamma \mathbb{E}_{s' \sim P_{s,a}} \left[ v_2 \left( s', \frac{r(s,a)+z}{\gamma} \right) \right] \right\} \\
&\leq \max_{a \in \mathcal{A}} \left\{ \tilde{r}(s,z,a) + \gamma \mathbb{E}_{s' \sim P_{s,a}} \left[ v_1 \left( s', \frac{r(s,a)+z}{\gamma} \right) \right] \right. \\
&\quad \left. - \tilde{r}(s,z,a) + \gamma \mathbb{E}_{s' \sim P_{s,a}} \left[ v_2 \left( s', \frac{r(s,a)+z}{\gamma} \right) \right] \right\} \\
&= \max_{a \in \mathcal{A}} \gamma \mathbb{E}_{s' \sim P_{s,a}} \left[ v_1 \left( s', \frac{r(s,a)+z}{\gamma} \right) - v_2 \left( s', \frac{r(s,a)+z}{\gamma} \right) \right] \\
&\leq \max_{a \in \mathcal{A}} \gamma \mathbb{E}_{s' \sim P_{s,a}} \left[ \left| v_1(s', \frac{r(s,a)+z}{\gamma}) - v_2 \left( s', \frac{r(s,a)+z}{\gamma} \right) \right| \right] \\
&\leq \max_{a \in \mathcal{A}} \gamma \mathbb{E}_{s' \sim P_{s,a}} [\|v_1 - v_2\|_\infty] = \gamma \|v_1 - v_2\|_\infty.
\end{aligned}
$$

This also similarly applies to:

$$
[\bar{T}v_2](s,z) - [\bar{T}v_1](s,z) \leq \gamma \|v_1 - v_2\|_\infty,
$$

hence $\|\bar{T}v_2 - \bar{T}v_1\|_\infty \leq \gamma \|v_1 - v_2\|_\infty$. $\qquad\square$

**Lemma B.2.** *The optimal value $\bar{v}^*$ is a fixed point of operator $\bar{T}$.*

*Proof.* Recall that $\tilde{r}(s,z,a) = z_- - (r(s,a)+z)_- = \min(0, r(s,a)+z) - \min(0,z)$. By definition of $\bar{T}$, for all $(s,z) \in \bar{\mathcal{S}}$, one can verify that

$$
\begin{aligned}
[\bar{T}\bar{v}^*](s,z) &= \max_{a \in \mathcal{A}} \left\{ \tilde{r}(s,z,a) + \gamma \mathbb{E}_{s' \sim P_{s,a}} \left[ \bar{v}^* \left( s', \frac{r(s,a)+z}{\gamma} \right) \right] \right\} \\
&= \max_{a \in \mathcal{A}} \left\{ \tilde{r}(s,z,a) + \gamma \mathbb{E}_{s' \sim P_{s,a}} \left[ \max_{\boldsymbol{\pi} \in \Pi^H} \mathbb{E}_{\tau \sim \mathbb{P}_{s'}^{\boldsymbol{\pi}}} \left[ \min \left( 0, R(\tau) + \frac{r(s,a)+z}{\gamma} \right) \right] \right. \right. \\
&\quad \left. \left. - \min \left( 0, \frac{r(s,a)+z}{\gamma} \right) \right] \right\} \\
&= \max_{a \in \mathcal{A}} \left\{ \min(0, r(s,a)+z) - \min(0,z) + \right. \\
&\quad \mathbb{E}_{s' \sim P_{s,a}} \left[ \max_{\boldsymbol{\pi} \in \Pi^H} \mathbb{E}_{\tau \sim \mathbb{P}_{s'}^{\boldsymbol{\pi}}} [\min(0, \gamma R(\tau) + r(s,a) + z)] - \gamma \min \left( 0, \frac{r(s,a)+z}{\gamma} \right) \right] \right\} \\
&= \max_{a \in \mathcal{A}} \left\{ \mathbb{E}_{s' \sim P_{s,a}} \left[ \max_{\boldsymbol{\pi} \in \Pi^H} \mathbb{E}_{\tau \sim \mathbb{P}_{s'}^{\boldsymbol{\pi}}} [\min(0, \gamma R(\tau) + r(s,a) + z)] \right] - \min(0,z) \right\} \\
&= \max_{\boldsymbol{\pi} \in \Pi^H} \mathbb{E}_{\tau \sim \mathbb{P}_s^{\boldsymbol{\pi}}} [\min(0, R(\tau) + z)] - \min(0,z) \\
&= \bar{v}^*(s,z).
\end{aligned}
$$

$\qquad\square$

**Lemma B.3.** *For all $(s,z) \in \bar{\mathcal{S}}$, it holds that $\bar{v}^*(s,z) = \bar{v}^*(s, \mathrm{p}(z))$.*

*Proof.* Recall that

$$\bar{v}^*(s, z) := \max_{\boldsymbol{\pi} \in \Pi^{\mathrm{H}}} \mathbb{E}_{\tau \sim \mathbb{P}_s^{\pi}} [-(R(\tau) + z)_- + z_-] = \max_{\boldsymbol{\pi} \in \Pi^{\mathrm{H}}} \mathbb{E}_{\tau \sim \mathbb{P}_s^{\pi}} [\min(0, R(\tau) + z)] - \min(0, z),$$

and that for all $\boldsymbol{\pi} \in \Pi^{\mathrm{H}}$ and initial state $s$, $R(\tau) \in [-r_\gamma, r_\gamma]$ almost surely.

We proceed case by case in each of the following three scenarios:

Case 1: $z \in [-r_\gamma, r_\gamma]$. By definition,

$$\mathrm{p}(z) = \max(-(1-\gamma)^{-1} r_{\max}, \min(z, (1-\gamma)^{-1} r_{\max})) = z,$$

so necessarily, $\bar{v}^*(s, \mathrm{p}(z)) = \bar{v}^*(s, z)$.

Case 2: $z \geq r_\gamma$. Then, $\mathrm{p}(z) = r_\gamma$ and

$$\begin{aligned}
\bar{v}^*(s, \mathrm{p}(z)) &= \bar{v}^*(s, r_\gamma) \\
&= \max_{\boldsymbol{\pi} \in \Pi^{\mathrm{H}}} \mathbb{E}_{\tau \sim \mathbb{P}_s^{\pi}} [\min(0, R(\tau) + r_\gamma)] - \min(0, r_\gamma) \\
&= 0 \\
&= \max_{\boldsymbol{\pi} \in \Pi^{\mathrm{H}}} \mathbb{E}_{\tau \sim \mathbb{P}_s^{\pi}} [\min(0, R(\tau) + z)] - \min(0, z) = \bar{v}^*(s, z),
\end{aligned}$$

where the second equality follows from $R(\tau) + r_\gamma \geq 0$ almost surely and $r_\gamma \geq 0$, and the last two inequalities hold due to $z \geq r_\gamma \geq 0$ and $R(\tau) + z \geq R(\tau) + r_\gamma \geq 0$ almost surely for all $\boldsymbol{\pi} \in \Pi^{\mathrm{H}}$.

Case 3: $z \leq -r_\gamma$. Then $\mathrm{p}(z) = -r_\gamma$ so that:

$$\begin{aligned}
\bar{v}^*(s, \mathrm{p}(z)) &= \bar{v}^*(s, -r_\gamma) \\
&= \max_{\boldsymbol{\pi} \in \Pi^{\mathrm{H}}} \mathbb{E}_{\tau \sim \mathbb{P}_s^{\pi}} [\min(0, R(\tau) - r_\gamma)] - \min(0, -r_\gamma) \\
&= \max_{\boldsymbol{\pi} \in \Pi^{\mathrm{H}}} \mathbb{E}_{\tau \sim \mathbb{P}_s^{\pi}} [R(\tau) - r_\gamma] + r_\gamma \\
&= \max_{\boldsymbol{\pi} \in \Pi^{\mathrm{H}}} \mathbb{E}_{\tau \sim \mathbb{P}_s^{\pi}} [R(\tau)] \\
&= \max_{\boldsymbol{\pi} \in \Pi^{\mathrm{H}}} \mathbb{E}_{\tau \sim \mathbb{P}_s^{\pi}} [R(\tau) + z] - z \\
&= \max_{\boldsymbol{\pi} \in \Pi^{\mathrm{H}}} \mathbb{E}_{\tau \sim \mathbb{P}_s^{\pi}} [\min(0, R(\tau) + z)] - \min(0, z) = \bar{v}^*(s, z),
\end{aligned}$$

where the second equality follows from $R(\tau) - r_\gamma \leq 0$ almost surely, and the last two equalities hold due to both $z \leq -r_\gamma \leq 0$ and $R(\tau) + z \leq R(\tau) - r_\gamma \leq 0$ almost surely for all $\pi \in \Pi$. $\qquad\square$

We can easily further confirm our claim that $\bar{v}^*$ is the only fixed point by arguing that any fixed point $\hat{v}$ of $\bar{T}$ must satisfy:

$$\|\hat{v} - \bar{v}^*\|_\infty = \|T\hat{v} - T\bar{v}^*\|_\infty \leq \gamma \|\hat{v} - \bar{v}^*\|_\infty \implies (1-\gamma)\|\hat{v} - \bar{v}^*\|_\infty \leq 0 \implies \|\hat{v} - \bar{v}^*\|_\infty \leq 0,$$

since $0 \leq \gamma < 1$.

## B.6. Proof of Proposition 5.2

Our proof relies on two useful lemmas.

### B.6.1. Two Useful Lemmas

We start with a simple Lemma about convex functions.

**Lemma B.4.** *Given a convex function $f(x)$, for all $\Delta \leq 0$ and $x_1 \leq x_2$, we have that:*

$$f(x_1) - f(x_1 + \Delta) \leq f(x_2) - f(x_2 + \Delta).$$

*Alternatively, if $f(x)$ is concave, then*

$$f(x_1) - f(x_1 + \Delta) \geq f(x_2) - f(x_2 + \Delta).$$

*Proof.* We start by showing that given $x_1 \leq x_2 \leq x_3$, we must have that:

$$s_1 := \frac{f(x_2) - f(x_1)}{x_2 - x_1} \leq s_2 := \frac{f(x_3) - f(x_1)}{x_3 - x_1} \leq s_3 := \frac{f(x_3) - f(x_2)}{x_3 - x_2}.$$

One can first use the convexity of $f(x)$ to confirm:

$$
\begin{aligned}
f(x_2) - f(x_1) &= f\left( \frac{x_2 - x_1}{x_3 - x_1} x_3 + \left(1 - \frac{x_2 - x_1}{x_3 - x_1}\right) x_1 \right) - f(x_1) \\
&\leq \frac{x_2 - x_1}{x_3 - x_1} f(x_3) + \left(1 - \frac{x_2 - x_1}{x_3 - x_1}\right) f(x_1) - f(x_1) \\
&= \frac{x_2 - x_1}{x_3 - x_1} (f(x_3) - f(x_1)).
\end{aligned}
$$

Hence, $s_1 \leq s_2$. Now, given that $s_2(x_3 - x_1) = s_1(x_2 - x_1) + s_3(x_3 - x_2)$, we must have:

$$
\begin{aligned}
s_2(x_3 - x_1) &= s_1(x_2 - x_1) + s_3(x_3 - x_2) \leq s_2(x_2 - x_1) + s_3(x_3 - x_2) \\
&\Rightarrow \; s_2(x_3 - x_2) \leq s_3(x_3 - x_2) \; \Rightarrow \; s_2 \leq s_3.
\end{aligned}
$$

Similarly

$$
\begin{aligned}
s_1(x_3 - x_1) &\leq s_2(x_3 - x_1) = s_1(x_2 - x_1) + s_3(x_3 - x_2) \\
&\Rightarrow \; s_1(x_3 - x_2) \leq s_3(x_3 - x_2) \; \Rightarrow \; s_1 \leq s_3.
\end{aligned}
$$

Now getting back at our claim, one can identify two cases. In a first case, we have that

$$x_1 - |\Delta| \leq x_1 \leq x_2 - |\Delta| \leq x_2.$$

Applying the ordering of secants established above twice over this sequence, we get:

$$\frac{f(x_1) - f(x_1 - |\Delta|)}{|\Delta|} \leq \frac{f(x_2 - |\Delta|) - f(x_1)}{x_2 - |\Delta| - x_1} \leq \frac{f(x_2) - f(x_2 - |\Delta|)}{|\Delta|},$$

concluding that $f(x_1) - f(x_1 - |\Delta|) \leq f(x_2) - f(x_2 - |\Delta|)$ since $|\Delta| \geq 0$.

Alternatively, in the second case we have $x_1 - |\Delta| \leq x_2 - |\Delta| \leq x_1 \leq x_2$. A similar argument leads to:

$$\frac{f(x_1) - f(x_1 - |\Delta|)}{|\Delta|} \leq \frac{f(x_1) - f(x_2 - |\Delta|)}{x_1 - x_2 + |\Delta|} \leq \frac{f(x_2) - f(x_2 - |\Delta|)}{|\Delta|}.$$

Hence, we have again that $f(x_1) - f(x_1 - |\Delta|) \leq f(x_2) - f(x_2 - |\Delta|)$.

The result for concave functions follows from $g(x) := -f(x)$ being convex, thus:

$$g(x_1) - g(x_1 + \Delta) \leq g(x_2) - g(x_2 + \Delta),$$

which implies that

$$-f(x_1) + f(x_1 + \Delta) \leq -f(x_2) + f(x_2 + \Delta),$$

and finally that

$$f(x_1) - f(x_1 + \Delta) \geq f(x_2) - f(x_2 + \Delta).$$

$\square$

We follow with a general lemma that will help verify that $\bar{T}^e$ preserves the monotonicity in $z$ of the value function.

**Lemma B.5.** *Let $f : \mathbb{R} \to \mathbb{R}$ be non-decreasing and $v \in \mathcal{L}_\infty(\bar{\mathcal{S}})$ be a non-decreasing function in $z$. Define for any $(s, z, a)$,*

$$g(s, z, a; v, f) := \left( \min(0, r(s,a) + z) - \min(0, z) \right) + \gamma \, \mathbb{E}_{s' \sim P_{s,a}} \left[ v\left( s', f\left( \frac{r(s,a) + z}{\gamma} \right) \right) \right].$$

*If Assumption 5.1 is satisfied, then for any $s \in \mathcal{S}$, $a \in \mathcal{A}$, $g(s, z, a, v, f)$ is non-decreasing in $z$.*

*Proof.* Fix $s \in \mathcal{S}$ and $a \in \mathcal{A}$, and let $z_1 \leq z_2$. For brevity write $r = r(s,a)$. Then

$$g(s, z_2, a, v, f) - g(s, z_1, a, v, f) = \Big( \min(0, r + z_2) - \min(0, z_2) \Big) - \Big( \min(0, r + z_1) - \min(0, z_1) \Big)$$
$$+ \gamma \mathbb{E}_{s' \sim P_{s,a}} \left[ v \left( s', f \left( \frac{r + z_2}{\gamma} \right) \right) - v \left( s', f \left( \frac{r + z_1}{\gamma} \right) \right) \right].$$

**(1)** The $\min$ function is concave. Since $r \leq 0$, the map $z \mapsto \min(z + r) - \min(z)$ is non-decreasing in $z$ (by Theorem B.4 applied to the concave function $\min$ with step $\Delta = r \leq 0$). Hence

$$\min(0, r + z_2) - \min(0, z_2) \;\geq\; \min(0, r + z_1) - \min(0, z_1).$$

**(2)** Since $z_1 \leq z_2$, we have $(r + z_1)/\gamma \leq (r + z_2)/\gamma$. Because $f$ is non-decreasing, $f((r + z_1)/\gamma) \leq f((r + z_2)/\gamma)$. This further implies by monotonicity of $v$ in $z$ that:

$$\gamma \mathbb{E}_{s' \sim P_{s,a}} \left[ v \left( s', f \left( \frac{r + z_2}{\gamma} \right) \right) \right] \geq \gamma \mathbb{E}_{s' \sim P_{s,a}} \left[ v \left( s', f \left( \frac{r + z_1}{\gamma} \right) \right) \right].$$

Combining (1) and (2) yields $g(s, z_2, a, v, f) \geq g(s, z_1, a, v, f)$, which proves the claim. $\qquad\square$

### B.6.2. MAIN PROOF

*Proof.* Since $\mathrm{l}(y) \leq y \leq \mathrm{u}(y)$ and $\mathrm{p}$ is non-decreasing, we have for all $y \in \mathbb{R}$:

$$\mathrm{p}(\mathrm{l}(y)) \leq \mathrm{p}(y) \leq \mathrm{p}(\mathrm{u}(y)). \tag{7}$$

Recall that:

$$[\bar{T}^{\mathrm{l}}\bar{v}](s, z) := [\bar{T}\bar{v}_{\mathrm{pol}}](s, z) = \max_a \left( \min(0, r(s,a) + z) - \min(0, z) \right) + \gamma \mathbb{E}_{s' \sim P_{s,a}} \left[ \bar{v} \left( s', \mathrm{p} \left( \mathrm{l} \left( \frac{r(s,a) + z}{\gamma} \right) \right) \right) \right]$$

$$[\bar{T}^{\mathrm{u}}\bar{v}](s, z) := [\bar{T}\bar{v}_{\mathrm{pou}}](s, z) = \max_a \left( \min(0, r(s,a) + z) - \min(0, z) \right) + \gamma \mathbb{E}_{s' \sim P_{s,a}} \left[ \bar{v} \left( s', \mathrm{p} \left( \mathrm{u} \left( \frac{r(s,a) + z}{\gamma} \right) \right) \right) \right],$$

to which we add:

$$[\bar{T}^{\mathrm{p}}\bar{v}](s, z) := [\bar{T}\bar{v}_{\mathrm{pop}}](s, z) = [\bar{T}\bar{v}_{\mathrm{p}}](s, z) = \max_a \left( \min(0, r(s,a) + z) - \min(0, z) \right) + \gamma \mathbb{E}_{s' \sim P_{s,a}} \left[ \bar{v} \left( s', \mathrm{p} \left( \frac{r(s,a) + z}{\gamma} \right) \right) \right].$$

Define

$$\bar{v}_k^{\mathrm{l}} := [\bar{T}^{\mathrm{l}}]^k \bar{v}, \qquad \bar{v}_k^{\mathrm{p}} := [\bar{T}^{\mathrm{p}}]^k \bar{v} \qquad \bar{v}_k^{\mathrm{u}} := [\bar{T}^{\mathrm{u}}]^k \bar{v}.$$

We will prove inductively that the following two properties hold:
(A) For all $k \geq 0$,
$$\bar{v}_k^{\mathrm{l}}(s, z) \;\leq\; \bar{v}_k^{\mathrm{p}}(s, z) \;\leq\; \bar{v}_k^{\mathrm{u}}(s, z) \qquad \forall (s, z) \in \bar{\mathcal{S}}.$$
(B) For every $s \in \mathcal{S}$, each of $\bar{v}_k^{\mathrm{l}}(s, z)$, $\bar{v}_k^{\mathrm{p}}(s, z)$, and $\bar{v}_k^{\mathrm{u}}(s, z)$ is non-decreasing in $z$.

**Base Case (k = 0):** Given that $\bar{v}_0^{\mathrm{l}} = \bar{v}_0^{\mathrm{p}} = \bar{v}_0^{\mathrm{u}} = \bar{v}$ and that $\bar{v}(s, z)$ is non-decreasing in $z$, (A) and (B) hold trivially for the base case.

**Induction:** Assume that $\bar{v}_k^{\mathrm{l}}(s, z) \;\leq\; \bar{v}_k^{\mathrm{p}}(s, z) \;\leq\; \bar{v}_k^{\mathrm{u}}(s, z) \;\; \forall (s, z) \in \bar{\mathcal{S}}$ and that all three functions are non-decreasing in $z$. We now show that properties (A) and (B) also apply at $k + 1$.

Starting with the envelope properties, we can see that for all $(s, z) \in \bar{\mathcal{S}}$:

$$\bar{v}_{k+1}^{\mathrm{u}}(s, z) = [\bar{T}^{\mathrm{u}}\bar{v}_k^{\mathrm{u}}](s, z) = \max_a \left( \min(0, r(s,a) + z) - \min(0, z) \right) + \gamma \mathbb{E}_{s' \sim P_{s,a}} \left[ \bar{v}_k^{\mathrm{u}} \left( s', \mathrm{p} \left( \mathrm{u} \left( \frac{r(s,a) + z}{\gamma} \right) \right) \right) \right]$$

$$\geq \max_a \left( \min(0, r(s,a) + z) - \min(0, z) \right) + \gamma \mathbb{E}_{s' \sim P_{s,a}} \left[ \bar{v}_k^{\mathrm{u}} \left( s', \mathrm{p} \left( \frac{r(s,a) + z}{\gamma} \right) \right) \right]$$

$$\geq \max_a \left( \min(0, r(s,a) + z) - \min(0, z) \right) + \gamma \mathbb{E}_{s' \sim P_{s,a}} \left[ \bar{v}_k^{\mathrm{p}} \left( s', \left( \frac{r(s,a) + z}{\gamma} \right) \right) \right] = [\bar{T}^{\mathrm{p}}\bar{v}_k^{\mathrm{p}}](s, z) = \bar{v}_{k+1}^{\mathrm{p}}(s, z),$$

where we first exploit (7) and the monotonicity of $v_l^{\mathrm{u}}$ with respect to $z$, and then $\bar{v}_k^{\mathrm{u}} \geq \bar{v}_k^{\mathrm{p}}$.

A similar argument applies for $\bar{v}_{k+1}^{\mathrm{l}}$. Namely,

$$\bar{v}_k^{\mathrm{l}}(s, z) = \max_a \left( \min(0, r(s, a) + z) - \min(0, z) \right) + \gamma \mathbb{E}_{s' \sim P_{s,a}} \left[ \bar{v}_k^{\mathrm{l}} \left( s', \mathrm{p} \left( 1 \left( \frac{r(s, a) + z}{\gamma} \right) \right) \right) \right]$$

$$\leq \max_a \left( \min(0, r(s, a) + z) - \min(0, z) \right) + \gamma \mathbb{E}_{s' \sim P_{s,a}} \left[ \bar{v}_k^{\mathrm{p}} \left( s', \mathrm{p} \left( \frac{r(s, a) + z}{\gamma} \right) \right) \right]$$

$$= [\bar{T}^{\mathrm{p}} \bar{v}_k^{\mathrm{p}}](s, z) = \bar{v}_{k+1}^{\mathrm{p}}(s, z).$$

Thus we have that for $k + 1$ property (A) holds.

We will now show that property (B) also holds at $k+1$ using Lemma (B.5). Namely, since $\bar{v}_{k+1}^{\mathrm{u}}$ and $\mathrm{p} \circ \mathrm{u}$ are non-decreasing, we have that:

$$\bar{v}_{k+1}^{\mathrm{u}}(s, z_2) - \bar{v}_{k+1}^{\mathrm{u}}(s, z_1) = \max_{a_2} g(s, z_2, a_2; \bar{v}_k^{\mathrm{u}}, \mathrm{p} \circ \mathrm{u}) - \max_{a_1} g(s, z_1, a_1; \bar{v}_k^{\mathrm{u}}, \mathrm{p} \circ \mathrm{u})$$

$$\geq \max_{a_2} g(s, z_2, a_2; \bar{v}_k^{\mathrm{u}}, \mathrm{p} \circ \mathrm{u}) - g(s, z_1, a_2; \bar{v}_k^{\mathrm{u}}, \mathrm{p} \circ \mathrm{u}) \geq 0.$$

Similarly for $\bar{v}_{k+1}^{\mathrm{l}}$,

$$\bar{v}_{k+1}^{\mathrm{l}}(s, z_2) - \bar{v}_{k+1}^{\mathrm{l}}(s, z_1) = \max_{a_2} g(s, z_2, a_2; \bar{v}_k^{\mathrm{l}}, \mathrm{p} \circ \mathrm{l}) - \max_{a_1} g(s, z_1, a_1; \bar{v}_k^{\mathrm{l}}, \mathrm{p} \circ \mathrm{l})$$

$$\geq \max_{a_2} g(s, z_2, a_2; \bar{v}_k^{\mathrm{l}}, \mathrm{p} \circ \mathrm{l}) - g(s, z_1, a_2; \bar{v}_k^{\mathrm{l}}, \mathrm{p} \circ \mathrm{l}) \geq 0,$$

and for $\bar{v}_{k+1}^{\mathrm{p}}$:

$$\bar{v}_{k+1}^{\mathrm{p}}(s, z_2) - \bar{v}_{k+1}^{\mathrm{p}}(s, z_1) = \max_{a_2} g(s, z_2, a_2; \bar{v}_k^{\mathrm{p}}, \mathrm{p}) - \max_{a_1} g(s, z_1, a_1; \bar{v}_k^{\mathrm{p}}, \mathrm{p})$$

$$\geq \max_{a_2} g(s, z_2, a_2; \bar{v}_k^{\mathrm{p}}, \mathrm{p}) - g(s, z_1, a_2; \bar{v}_k^{\mathrm{p}}, \mathrm{p}) \geq 0.$$

This completes our proof. $\qquad\square$

### B.7. Proof of Theorem 5.3

#### B.7.1. USEFUL LEMMAS

We start with two simple lemmas that investigate the magnitude of difference $\mathrm{p}(\mathrm{e}(y)) - \mathrm{e}(y)$ when $\mathrm{e} \in \{\mathrm{l}, \mathrm{u}\}$ and the continuity of $\bar{v}^*(s, z)$ in $z$.

**Lemma B.6.** *Given the projection map* $\mathrm{p}(y) := \max(-r_\gamma, \min(r_\gamma, y))$ *and any function* $f : \mathbb{R} \to \mathbb{R}$, *we have that* $\sup_{z \in \mathbb{R}} |\mathrm{p}(z) - \mathrm{p}(f(z))| \leq \sup_{z \in \mathbb{R}} |z - e(z)|$.

*Proof.* We first show that $\mathrm{p}$ is 1-Lipschitz continuous. Namely, for any $x, y \in \mathbb{R}$, the projection map $\mathrm{p}$ satisfies

$$|\mathrm{p}(x) - \mathrm{p}(y)| \leq |x - y|.$$

Assuming without loss of generality that $x \geq y$, we consider the six possible cases.

- If $x \geq y \geq r_\gamma$, then $|\mathrm{p}(x) - \mathrm{p}(y)| = |r_\gamma - r_\gamma| = 0 \leq |x - y|$.

- If $x \geq r_\gamma \geq y \geq -r_\gamma$, then $|\mathrm{p}(x) - \mathrm{p}(y)| = \mathrm{p}(x) - \mathrm{p}(y) = r_\gamma - y \leq x - y = |x - y|$.

- If $x \geq r_\gamma$ and $y \leq -r_\gamma$, then $|\mathrm{p}(x) - \mathrm{p}(y)| = |r_\gamma - (-r_\gamma)| = 2r_\gamma \leq x - y = |x - y|$.

- If $r_\gamma \geq x \geq y \geq -r_\gamma$, then $|\mathrm{p}(x) - \mathrm{p}(y)| = |x - y|$.

- If $r_\gamma \geq x \geq -r_\gamma \geq y$, then $|\mathrm{p}(x) - \mathrm{p}(y)| = \mathrm{p}(x) - \mathrm{p}(y) = x - (-r_\gamma) = x + r_\gamma \leq x - y = |x - y|$.

- If $-r_\gamma \geq x \geq y$, then $|\mathrm{p}(x) - \mathrm{p}(y)| = |-r_\gamma - (-r_\gamma)| = 0 \leq |x - y|$.

We thus conclude that $\sup_{z \in \mathbb{R}} |\mathrm{p}(z) - \mathrm{p}(f(z))| \leq |z - \mathrm{e}(z)| \leq \sup_{z \in \mathbb{R}} |z - f(z)|$. $\qquad\square$

We follow with a Lemma to show that for any state $s$, the optimal value function $\bar{v}^*(s, z) \in \mathcal{L}_\infty$ is 1-Lipschitz in $z$.

**Lemma B.7.** *The optimal value function $\bar{v}^*(s, z)$ is 1-Lipschitz in $z$.*

*Proof.* Recall the definition of $\bar{v}^*$:

$$\bar{v}^*(s, z) := \max_{\boldsymbol{\pi} \in \Pi^{\mathrm{H}}} \mathbb{E}_{\tau \sim \mathbb{P}_s^\pi}[f(R(\tau), z)],$$

with

$$f(y, z) := \min(0, y + z) - \min(0, z) = \begin{cases} 0 & \text{if } z \geq \max(-y, 0) \\ y + z & \text{if } 0 \leq z \leq -y \\ -z & \text{if } -y \leq z \leq 0 \\ y & \text{if } z \leq \min(-y, 0). \end{cases}$$

The function $f(y, z)$ is continuous in $z$ with one-sided derivatives in the set $\{-1, 0, 1\}$. We conclude that the function is Lipschitz continuous with Lipschitz constant of 1.

Next, we can straightforwardly check that for all $s \in \bar{\mathcal{S}}$ and $z_1, z_2 \in \mathbb{R}$:

$$\bar{v}^*(s, z_1) - \bar{v}^*(s, z_2) = \max_{\boldsymbol{\pi} \in \Pi^{\mathrm{H}}} \mathbb{E}_{\tau \sim \mathbb{P}_s^\pi}[f(R(\tau), z_1)] - \max_{\boldsymbol{\pi} \in \Pi^{\mathrm{H}}} \mathbb{E}_{\tau \sim \mathbb{P}_s^\pi}[f(R(\tau), z_2)]$$

$$\leq \max_{\boldsymbol{\pi} \in \Pi^{\mathrm{H}}} \mathbb{E}_{\tau \sim \mathbb{P}_s^\pi}[f(R(\tau), z_1)] - \mathbb{E}_{\tau \sim \mathbb{P}_s^\pi}[f(R(\tau), z_2)] = \max_{\boldsymbol{\pi} \in \Pi^{\mathrm{H}}} \mathbb{E}_{\tau \sim \mathbb{P}_s^\pi}[f(R(\tau), z_1) - f(R(\tau), z_2)]$$

$$\leq \max_{\boldsymbol{\pi} \in \Pi^{\mathrm{H}}} \mathbb{E}_{\tau \sim \mathbb{P}_s^\pi}[|z_1 - z_2|] = |z_1 - z_2|.$$

A similar property can be derived for $\bar{v}^*(s, z_2) - \bar{v}^*(s, z_1) \leq |z_1 - z_2|$. This completes our proof.

$\qquad\square$

### B.7.2. MAIN PROOF

*Proof.* The proof of contraction is very similar to that of Theorem 4.2 shown in Appendix B.5. In fact, given any mapping $f : \mathbb{R} \to \mathbb{R}$ and letting

$$[\bar{T}^f \bar{v}](s, z) := \max_a \left( \min(0, r(s, a) + z) - \min(0, z) \right) + \gamma \mathbb{E}_{s' \sim P_{s,a}} \left[ \bar{v} \left( s', p \left( f \left( \frac{r(s, a) + z}{\gamma} \right) \right) \right) \right],$$

one can straightforwardly show that for all $v_1, v_2 \in \mathcal{L}_\infty(\bar{\mathcal{S}})$ we have:

$$[\bar{T}^f v_1](s, z) - [\bar{T}^f v_2](s, z) = \max_{a \in \mathcal{A}} \left\{ \tilde{r}(s, z, a) + \gamma \mathbb{E}_s^a \left[ v_1 \left( s', p \left( f \left( \frac{r(s, a) + z}{\gamma} \right) \right) \right) \right] \right\}$$

$$- \max_{a \in \mathcal{A}} \left\{ \tilde{r}(s, z, a) + \gamma \mathbb{E}_s^a \left[ v_2 \left( s', p \left( f \left( \frac{r(s, a) + z}{\gamma} \right) \right) \right) \right] \right\}$$

$$\leq \max_{a \in \mathcal{A}} \left\{ \tilde{r}(s, z, a) + \gamma \mathbb{E}_s^a \left[ v_1 \left( s', p \left( f \left( \frac{r(s, a) + z}{\gamma} \right) \right) \right) \right] \right.$$

$$\left. - \tilde{r}(s, z, a) + \gamma \mathbb{E}_s^a \left[ v_2 \left( s', p \left( f \left( \frac{r(s, a) + z}{\gamma} \right) \right) \right) \right] \right\}$$

$$= \max_{a \in \mathcal{A}} \gamma \mathbb{E}_s^a \left[ v_1 \left( s', p \left( f \left( \frac{r(s, a) + z}{\gamma} \right) \right) \right) - v_2 \left( s', p \left( f \left( \frac{r(s, a) + z}{\gamma} \right) \right) \right) \right]$$

$$\leq \max_{a \in \mathcal{A}} \gamma \mathbb{E}_s^a [\|v_1 - v_2\|_\infty] = \gamma \|v_1 - v_2\|_\infty.$$

and similarly $[\bar{T}^f v_2](s, z) - [\bar{T}^f v_1](s, z) \leq \gamma \|v_1 - v_2\|_\infty$. Thus confirming the sup-norm $\gamma$-contraction of $\bar{T}^f$ for all $f \in \{l, p, u\}$.

By Banach Fixed Point theorem, we have that each of the $\bar{T}^f$ operators with $f \in \{\mathrm{l}, \mathrm{p}, \mathrm{u}\}$ respectively converges to a unique fixed point, namely $\bar{v}^{*,\mathrm{l}}$, $\bar{v}^{*,\mathrm{p}}$, and $\bar{v}^{*,\mathrm{u}}$. Moreover, Proposition 5.2 tells us that, since the zero function $\bar{v}_0(s,z) := 0$ is non-decreasing in $z$, for all $(s,z) \in \bar{\mathcal{S}}$, we can conclude that:

$$\bar{v}^{*,\mathrm{l}}(s,z) = \lim_{k\to\infty}[\bar{T}^{\mathrm{l}}]^k\bar{v}_0(s,z) \leq \lim_{k\to\infty}[\bar{T}^{\mathrm{p}}]^k\bar{v}_0(s,z) = \bar{v}^{*,\mathrm{p}} \leq \lim_{k\to\infty}[\bar{T}^{\mathrm{u}}]^k\bar{v}_0(s,z) = \bar{v}^{*,\mathrm{u}}.$$

Furthermore, for any $\mathrm{e} \in \{\mathrm{l}, \mathrm{p}, \mathrm{u}\}$, since each $[\bar{T}^{\mathrm{e}}]^k\bar{v}_0(s,z)$ is non-decreasing in $z$ for all $k \geq 0$ (see Proposition 5.2), it must be that $\bar{v}^{*,\mathrm{e}}(s,z)$ is non-decreasing in $z$.

Finally, Lemma B.3 allows us to verify that $\bar{v}^{*,\mathrm{p}} = \bar{v}^*$ using:

$$\bar{v}^{*,\mathrm{p}}(s,z) = \lim_{k\to\infty}[\bar{T}^{\mathrm{p}}]^k\bar{v}^*(s,z) = \bar{v}^*(s,z),$$

since $[\bar{T}^{\mathrm{p}}\bar{v}^*](s,z) = [\bar{T}\bar{v}_{\mathrm{p}}^*](s,z) = \bar{v}^*(s,z)$.

We are left with showing that $\|\bar{v}^* - \bar{v}_{\mathrm{poe}}^{*,\mathrm{e}}\|_\infty \leq \bar{\Delta}_{\mathrm{e}}$ for $\mathrm{e} \in \{\mathrm{l}, \mathrm{u}\}$. Namely, we have that:

$$\|\bar{v}^{*,\mathrm{e}} - \bar{v}^*\|_\infty = \lim_{K\to\infty}\|[\bar{T}^{\mathrm{e}}]^K\bar{v}^* - \bar{v}^*\|_\infty = \lim_{K\to\infty}\|\sum_{k=0}^{K-1}[\bar{T}^{\mathrm{e}}]^{k+1}\bar{v}^* - [\bar{T}^{\mathrm{e}}]^k\bar{v}^*\|_\infty$$

$$\leq \lim_{K\to\infty}\sum_{k=0}^{K-1}\|[\bar{T}^{\mathrm{e}}]^{k+1}\bar{v}^* - [\bar{T}^{\mathrm{e}}]^k\bar{v}^*\|_\infty \leq \lim_{K\to\infty}\sum_{k=0}^{K-1}\gamma^{K-1}\|\bar{T}^{\mathrm{e}}\bar{v}^* - \bar{v}^*\|_\infty$$

$$\leq \frac{1}{1-\gamma}\|\bar{T}^{\mathrm{e}}\bar{v}^* - \bar{v}^*\|_\infty$$

$$\leq \frac{1}{1-\gamma}\left(\|\bar{T}^{\mathrm{e}}\bar{v}^* - \bar{T}\bar{v}^*\|_\infty + \|\bar{T}\bar{v}^* - \bar{v}^*\|_\infty\right)$$

$$\leq \frac{1}{1-\gamma}\gamma\sup_{z\in\mathbb{R}}|\mathrm{e}(z) - z| = \bar{\Delta}_{\mathrm{e}},$$

where we exploited heavily the fact that $\bar{v}^{*,\mathrm{e}}$ is the unique fixed point of the $\gamma$-contraction $\bar{T}^{\mathrm{e}}$, the second and fourth inequalities apply the triangle inequality, whereas the last inequality needs more detailed explanation. Precisely, it follows from the fact that for all $(z,s) \in \bar{\mathcal{S}}$:

$$|[\bar{T}\bar{v}^*](s,z) - [\bar{T}^{\mathrm{e}}\bar{v}^*](s,z)| = |\bar{T}\bar{v}_{\mathrm{p}}^*(s,z) - \bar{T}^{\mathrm{e}}\bar{v}^*(s,z)|$$

$$= \left|\max_a\left\{\tilde{r}(s,z,a) + \gamma\mathbb{E}_{s'\sim P_{s,a}}\left[\bar{v}^*\left(s', \mathrm{p}\left(\frac{r(s,a)+z}{\gamma}\right)\right)\right]\right\}\right.$$

$$\left. - \max_a\left\{\tilde{r}(s,z,a) + \gamma\mathbb{E}_{s'\sim P_{s,a}}\left[\bar{v}^*\left(s', \mathrm{p}\left(\mathrm{e}\left(\frac{r(s,a)+z}{\gamma}\right)\right)\right)\right]\right\}\right|$$

$$\leq \max_a\left|\gamma\mathbb{E}_{s'\sim P_{s,a}}\left[\bar{v}^*\left(s', \mathrm{p}\left(\frac{r(s,a)+z}{\gamma}\right)\right) - \bar{v}^*\left(s', \mathrm{p}\left(\mathrm{e}\left(\frac{r(s,a)+z}{\gamma}\right)\right)\right)\right]\right|$$

$$\leq \gamma\max_a\mathbb{E}_{s'\sim P_{s,a}}\left[\left|\bar{v}^*\left(s', \mathrm{p}\left(\frac{r(s,a)+z}{\gamma}\right)\right) - \bar{v}^*\left(s', \mathrm{p}\left(\mathrm{e}\left(\frac{r(s,a)+z}{\gamma}\right)\right)\right)\right|\right]$$

$$\leq \gamma\max_a\mathbb{E}_{s'\sim P_{s,a}}\left[\left|\mathrm{p}\left(\frac{r(s,a)+z}{\gamma}\right) - \mathrm{p}\left(\mathrm{e}\left(\frac{r(s,a)+z}{\gamma}\right)\right)\right|\right]$$

$$\leq \gamma\sup_{z\in\mathbb{R}}|\mathrm{p}(z) - \mathrm{p}(\mathrm{e}(z))| \leq \gamma\sup_{z\in\mathbb{R}}|z - \mathrm{e}(z)|,$$

where we first exploit the fact that $\bar{v}^* = \bar{v}_{\mathrm{p}}^*$ (see Theorem 4.2), then the first inequality follows from $|\max_a h_1(a) - \max_a h_2(a)| \leq \max_a |h_1(a) - h_2(a)|$ for any $h_1, h_2 : \mathcal{A} \to \mathbb{R}$, the next inequality follows from Jensen inequality, and we are left with applying Lemma B.6 and B.7.

$\square$

## B.8. Proof of Theorem 5.4

*Proof.* We start with showing that:

$$\Psi^* - \bar{\Delta}_l/\alpha \leq \Psi^l \leq \Psi^* \leq \Psi^u \leq \Psi^* + \alpha\bar{\Delta}_u/\alpha,$$

which follows naturally from Theorem 5.3. Namely,

$$
\begin{aligned}
\Psi^* - \bar{\Delta}_l/\alpha &= \sup_z \frac{1}{\alpha}(\bar{v}^*(\bar{s}_0, z) + \min(0, z)) - z - \bar{\Delta}_l/\alpha \\
&= \sup_z \frac{1}{\alpha}(\bar{v}^*(\bar{s}_0, z) - \bar{\Delta}_l + \min(0, z)) - z \\
&\leq \Psi^l := \sup_z \frac{1}{\alpha}(\bar{v}^{l,*}(\bar{s}_0, z) + \min(0, z)) - z \\
&\leq \sup_z \frac{1}{\alpha}(\bar{v}^*(\bar{s}_0, z) + \min(0, z)) - z \\
&= \Psi^* := \max_{\pi \in \Pi^H} \mathrm{CVaR}_\alpha^{\bar{s}_0, \pi}[R(\tau)] \\
&\leq \Psi^{*,u} := \sup_z \frac{1}{\alpha}(\bar{v}^{u,*}(\bar{s}_0, z) + \min(0, z)) - z \\
&\leq \sup_z \frac{1}{\alpha}(\bar{v}^*(\bar{s}_0, z) + \bar{\Delta}_u + \min(0, z)) - z \\
&= \Psi^* + \bar{\Delta}_u/\alpha.
\end{aligned}
$$

Considering the performance of the history dependent policy $\bar{\pi}(\cdot, z_1^*)$, it is enough to show that for all $s \in \mathcal{S}$:

$$\mathbb{E}_{\tau \sim \mathbb{P}_s^{\bar{\pi}(\cdot, z_1^*)}}[-(R(\tau) + z_1^*)_- + (z_1^*)_-] \geq \bar{v}^{l*}(s, z_1^*). \tag{8}$$

This would then easily lead to:

$$
\begin{aligned}
\mathrm{CVaR}_\alpha^{\bar{s}_0, \bar{\pi}(\cdot, z_1^*)}[R(\tau)] &= \sup_z -z - \frac{z_-}{\alpha} + \frac{1}{\alpha}\mathbb{E}_{\tau \sim \mathbb{P}_{\bar{s}_0}^{\bar{\pi}(\cdot, z_1^*)}}[-(R(\tau) + z)_- + z_-] \\
&\geq -z_1^* - \frac{(z_1^*)_-}{\alpha} + \frac{1}{\alpha}\mathbb{E}_{\tau \sim \mathbb{P}_{\bar{s}_0}^{\bar{\pi}(\cdot, z_1^*)}}[-(R(\tau) + z_1^*)_- + (z_1^*)_-] \\
&\geq -z_1^* - \frac{(z_1^*)_-}{\alpha} + \frac{1}{\alpha}\bar{v}^{l*}(\bar{s}_0, z_1^*) = \Psi^l.
\end{aligned}
$$

To show that equation (8) holds, we can study the value function:

$$\hat{v}_k^\dagger(s, z) := \mathbb{E}_{\tau \sim \mathbb{P}_s^{\bar{\pi}(\cdot, z_1^*)}}\left[-\left(\sum_{t=0}^{k-1} \gamma^t r(s_t, a_t) + z\right)_- + z_-\right],$$

and Bellman operator:

$$[\hat{T}^\dagger v](s, z) := \tilde{r}(s, \tilde{\pi}^*(s, z)) + \gamma\mathbb{E}_{s' \sim P_{s, \tilde{\pi}^*(s, z)}}\left[v\left(s', \mathrm{p}\left(\mathrm{l}\left(\frac{r(s, \tilde{\pi}^*(s, z)) + z}{\gamma}\right)\right)\right)\right],$$

for which we can prove three properties.

**First property:** $\hat{T}^\dagger$ is a sup-norm contraction in $\mathcal{L}_\infty(\bar{\mathcal{S}})$. This follows using the same steps as in the proof of Theorem 4.2 shown in Appendix B.5.

**Second property:** We have that for all $k \geq 0$:

$$\hat{v}_k^\dagger(s, z) \geq [[\hat{T}^\dagger]^k \bar{v}_0](s, z), \, \forall(s, z) \in \bar{\mathcal{S}}$$

for $\bar{v}_0(s, z) := 0$. Namely, we can show this inductively starting from $k = 0$ where

$$\hat{v}_0^\dagger(s, z) := \mathbb{E}_{\tau \sim \mathbb{P}_s^{\bar{\pi}(\cdot, z_1^*)}} \left[ -\left( \sum_{t=0}^{0-1} \gamma^t r(s_t, a_t) + z \right)_- + z_- \right] = -z_- + z_- = \hat{v}_0^\dagger(s, z).$$

We follow with at $k + 1$:

$$\hat{v}_{k+1}^\dagger(s, z) = \mathbb{E}_{\tau \sim \mathbb{P}_s^{\bar{\pi}(\cdot, z_1^*)}} \left[ \min\left( 0, z + \sum_{t=0}^k \gamma^t r(s_t, a_t) \right) - \min(0, z) \right]$$

$$= \min(0, z + r(s, \tilde{\pi}^*(s, z))) - \min(0, z)$$

$$+ \mathbb{E}_{\tau \sim \mathbb{P}_s^{\bar{\pi}(\cdot, z_1^*)}} \left[ \min\left( 0, z + \sum_{t=0}^k \gamma^t r(s_t, a_t) \right) \right] - \min(0, z + r(s, \tilde{\pi}^*(s, z)))$$

$$= \min(0, z + r(s, \tilde{\pi}^*(s, z))) - \min(0, z)$$

$$+ \gamma \left( \mathbb{E}_{\tau \sim \mathbb{P}_s^{\bar{\pi}(\cdot, z_1^*)}} \left[ \min\left( 0, \frac{z + r(s, \tilde{\pi}^*(s, z))}{\gamma} + \sum_{t=1}^k \gamma^{t-1} r(s_t, a_t) \right) \right] - \min\left( 0, \frac{z + r(s, \tilde{\pi}^*(s, z))}{\gamma} \right) \right)$$

$$= \min(0, z + r(s, \tilde{\pi}^*(s, z))) - \min(0, z)$$

$$+ \gamma \left( \mathbb{E}_{\tau \sim \mathbb{P}_s^{\bar{\pi}(\cdot, z_1^*)}} \left[ \min\left( 0, \mathrm{p}\left( \frac{z + r(s, \tilde{\pi}^*(s, z))}{\gamma} \right) + \sum_{t=1}^k \gamma^{t-1} r(s_t, a_t) \right) \right] - \min\left( 0, \mathrm{p}\left( \frac{z + r(s, \tilde{\pi}^*(s, z))}{\gamma} \right) \right) \right)$$

$$\geq \min(0, z + r(s, \tilde{\pi}^*(s, z))) - \min(0, z)$$

$$+ \gamma \left( \mathbb{E}_{\tau \sim \mathbb{P}_s^{\bar{\pi}(\cdot, z_1^*)}} \left[ \min\left( 0, \mathrm{p}(\mathrm{l}(\frac{z + r(s, \tilde{\pi}^*(s, z))}{\gamma})) + \sum_{t=1}^k \gamma^{t-1} r(s_t, a_t) \right) \right] - \min\left( 0, \mathrm{p}\left( \mathrm{l}\left( \frac{z + r(s, \tilde{\pi}^*(s, z))}{\gamma} \right) \right) \right) \right)$$

$$= \min(0, z + r(s, \tilde{\pi}^*(s, z))) - \min(0, z)$$

$$+ \gamma \left( \mathbb{E}_{s' \sim P_{s, \tilde{\pi}^*(s, z)}} \left[ \hat{v}_k^\dagger \left( s', \mathrm{p}\left( \mathrm{l}\left( \frac{z + r(s, \tilde{\pi}^*(s, z))}{\gamma} \right) \right) \right) \right] \right)$$

$$= [\hat{T}^\dagger \hat{v}_k](s, z) \geq [\hat{T}^\dagger [\hat{T}^\dagger]^k \bar{v}_0](s, z) = [[\hat{T}^\dagger]^{k+1} \bar{v}_0](s, z),$$

where the third equality follows from:

$$\min\left( 0, y + \sum_{t=1}^k \gamma^{t-1} r(s_t, a_t) \right) - \min(0, y) = \min\left( 0, \mathcal{P}(y) + \sum_{t=1}^k \gamma^{t-1} r(s_t, a_t) \right) - \min(0, \mathcal{P}(y))$$

since $|\sum_{t=1}^k \gamma^{t-1} r(s_t, a_t)| \leq \frac{r_{\max}}{1-\gamma} \leq r_\gamma$. Furthermore, the first inequality comes from Lemma B.4 and the envelope property $\mathrm{l}(y) \leq y$ thus when we let $x_2 := \mathcal{P}((z + r(s, \tilde{\pi}^*(s, z)))/\gamma) \geq \mathcal{P}(\mathrm{l}((z + r(s, \tilde{\pi}^*(s, z)))/\gamma)) =: x_1$ and $\Delta := \sum_{t=1}^k \gamma^{t-1} r(s_t, a_t) \leq 0$, then we have that:

$$\min(0, x_2 + \Delta) - \min(0, x_2) \geq \min(0, x_1 + \Delta) - \min(0, x_1),$$

where $\min(0, y)$ is concave in $y$.

**Third property:** We have that :

$$[\hat{T}^\dagger \bar{v}^{l*}](s, z) = (\min(0, r(s, \tilde{\pi}^*(s, z)) + z) - \min(0, z)) + \gamma \mathbb{E}_{s' \sim P_{s, \tilde{\pi}^*(s, z)}} \left[ \bar{v}^{l*} \left( s', \mathrm{p}\left( \mathrm{l}\left( \frac{z + r(s, \tilde{\pi}^*(s, z))}{\gamma} \right) \right) \right) \right]$$

$$= \max_a \left( \min(0, r(s, a) + z) - \min(0, z) \right) + \gamma \mathbb{E}_{s' \sim P_{s, a}} \left[ \bar{v}^{l*} \left( s', \mathrm{p}\left( \mathrm{l}\left( \frac{z + r(s, a)}{\gamma} \right) \right) \right) \right]$$

$$= [\bar{T}^l \bar{v}^{l*}](s, z) = \bar{v}^{l*}(s, z),$$

or in other words $\bar{v}^{l*}$ is a fixed point of $\hat{T}^\dagger$. So that, since $\hat{T}^\dagger$ is a $\gamma$-contraction (see the first property derived), we have that $[\hat{T}^\dagger]^k \bar{v}_0$, with $\bar{v}_0 := 0$, converges to $\bar{v}^{l*}$. Hence, we can conclude that

$$\mathbb{E}_{\tau \sim \mathbb{P}_s^{\bar{\pi}(\cdot, z_1^*)}} \left[ \min\left(0, z + \sum_{t=0}^{k} \gamma^t r(s_t, a_t)\right) - \min(0, z) \right] = \lim_{k \to \infty} \hat{v}_k^\dagger(s, z) \geq \lim_{k \to \infty} [[\hat{T}^\dagger]^k \bar{v}_0](s, w) = \bar{v}^{l*}(s, z).$$

This supports the claim made in equation (8) thus completes our proof. $\qquad \square$

## B.9. Proof of Proposition 5.5

*Proof.* For clarity purposes, we first redefine the Bellman operators $\bar{T}^e : \mathcal{L}_\infty(\mathcal{S} \times \mathcal{Z}_\Delta) \to \mathcal{L}_\infty(\mathcal{S} \times \mathcal{Z}_\Delta)$ with $e \in \{l_\Delta, u_\Delta\}$. Specifically, for all $\hat{v} \in \mathcal{L}_\infty(\mathcal{S} \times \mathcal{Z}_\Delta)$ we let:

$$[\bar{T}^e \hat{v}](s, z) := \max_{a \in \mathcal{A}} \left\{ \tilde{r}(s, z, a) + \gamma \mathbb{E}_{s' \sim P_{s,a}} \left[ \hat{v}\left(s', p\left(e\left(\frac{r(s, a) + z}{\gamma}\right)\right)\right) \right] \right\}, \forall (s, z) \in \mathcal{Z}_\Delta,$$

which is well defined since one can confirm that

$$(s, z) \in \mathcal{S} \times \mathbb{R} \Rightarrow e\left(\frac{r(s, a) + z}{\gamma}\right) \in \{k\Delta : k \in \mathbb{Z}\} \Rightarrow p\left(e\left(\frac{r(s, a) + z}{\gamma}\right)\right) \in (\{k\Delta : k \in \mathbb{Z}\} \cap [-r_\gamma, r_\gamma]) \cup \{r_\gamma, r_\gamma\} \subseteq \mathcal{Z}_\Delta,$$

since $\{-r_\gamma, r_\gamma\} = \{-K\Delta, K\Delta\} \subseteq \{k\Delta : k \in \mathbb{Z}\}$, hence $(s', p(e(\frac{r(s,a)+z}{\gamma})))$ is in the domain of $\hat{v}$.

One can further follow the same steps as in the proof of Theorem 5.3 to show that $\bar{T}^e$ is a $\gamma$-contraction in $\mathcal{L}_\infty(\mathcal{S} \times \mathcal{Z}_\Delta)$ and therefore admits a unique fixed point $\hat{v}^{*,e}$. Letting $\check{v}(s, z) := \bar{T}^e \hat{v}^{*,e}(s, z)$ for all $(s, z) \in \bar{\mathcal{S}}$, we can finally verify that for all $(s, z) \in \bar{S}$:

$$\begin{aligned}
[\bar{T}^e \check{v}](s, z) &= \max_{a \in \mathcal{A}} \left\{ \tilde{r}(s, z, a) + \gamma \mathbb{E}_{s' \sim P_{s,a}} \left[ \check{v}\left(s', p\left(e\left(\frac{r(s, a) + z}{\gamma}\right)\right)\right) \right] \right\} \\
&= \max_{a \in \mathcal{A}} \left\{ \tilde{r}(s, z, a) + \gamma \mathbb{E}_{s' \sim P_{s,a}} \left[ \bar{T}^e \hat{v}^{*,e}\left(s', p\left(e\left(\frac{r(s, a) + z}{\gamma}\right)\right)\right) \right] \right\} \\
&= \max_{a \in \mathcal{A}} \left\{ \tilde{r}(s, z, a) + \gamma \mathbb{E}_{s' \sim P_{s,a}} \left[ \hat{v}^{*,e}\left(s', p\left(e\left(\frac{r(s, a) + z}{\gamma}\right)\right)\right) \right] \right\} \\
&= [\bar{T}^e \hat{v}^{*,e}](s, z) = \check{v}(s, z),
\end{aligned}$$

where the third equality follows from the $\bar{T}^e$ on $\mathcal{L}_\infty(\bar{\mathcal{S}})$ reducing to the $\bar{T}^e$ on $\mathcal{L}_\infty(\mathcal{S} \times \mathcal{Z}_\Delta)$ since the term $\bar{T}^e \hat{v}^{*,e}(s', z)$ that appears in the expectation only gets evaluated on $z \in \mathcal{Z}_\Delta$. This confirms that $\check{v}(s, z)$ is a fixed point of $\bar{T}^e$ in $\mathcal{L}_\infty(\bar{\mathcal{S}})$ thus must be equal to $\bar{v}^{*,e}(s, z)$.

$\qquad \square$

## B.10. Proof of Proposition 6.1

*Proof.* Algorithm 1 iterates on the following Q-function Bellman update $\hat{q}_{k+1}^e := \bar{\mathfrak{T}}^e \hat{q}_k^e$ with

$$[\bar{\mathfrak{T}}^e \hat{q}_k^e](s, z, a) := \tilde{r}(s, z, a) + \gamma \sum_{s' \in \mathcal{S}} P(s'|s, a) \max_{a' \in \mathcal{A}} \hat{q}_k^e(s', p \circ e(\gamma^{-1}(r(s, a) + z)), a'), \forall (s, z, a) \in \mathcal{S} \times \mathcal{Z}_\Delta,$$

and $\hat{q}_0^e(s, z, a) := 0$, until the stopping criterion $\|\bar{q}_k^e - \bar{q}_{k-1}^e\|_\infty \leq \epsilon$ is met.

Similarly to $\bar{T}^e$, $\bar{\mathfrak{T}}^e$ is know to be a $\gamma$-contraction with respect to the sup-norm (see Bertsekas (1995)) with $\hat{q}^{*,e}$ as its unique fixed point in $\mathcal{L}_\infty(\mathcal{S} \times \mathcal{Z}_\Delta \times \mathcal{A})$. This implies that

$$\|\hat{q}^{*,e} - \bar{q}_k^e\|_\infty = \lim_{K \to \infty} \|[\mathfrak{T}^e]^K \bar{q}_k^e - \bar{q}_k^e\|_\infty = \lim_{K \to \infty} \|\sum_{k'=0}^{K-1} [\mathfrak{T}^e]^{k'+1} \bar{q}_k^e - [\mathfrak{T}^e]^{k'} \bar{q}_k^e\|_\infty \leq \lim_{K \to \infty} \sum_{k'=0}^{K-1} \|[\mathfrak{T}^e]^{k'+1} \bar{q}_k^e - [\mathfrak{T}^e]^{k'} \bar{q}_k^e\|_\infty$$

$$\leq \lim_{K \to \infty} \sum_{k'=0}^{K-1} \gamma^{k'} \|\mathfrak{T}^e \bar{q}_k^e - \bar{q}_k^e\|_\infty = \frac{1}{1 - \gamma} \|\mathfrak{T}^e \bar{q}_k^e - \bar{q}_k^e\|_\infty \leq \frac{\gamma}{1 - \gamma} \|\mathfrak{T}^e \bar{q}_{k-1}^e - \bar{q}_{k-1}^e\|_\infty \leq \frac{\gamma \epsilon}{1 - \gamma}.$$

$\qquad \square$

### B.11. Proof of Theorem 6.2

*Proof.* We start by recalling that the Q-update rule in Line 14 of Algorithm 2 takes the form:

$$\hat{q}^e_{k+1}(s, z, a) := \hat{q}^e_k(s, z, a)$$
$$+ \begin{cases} \beta(N_k(s_k, a_k))\Big(\tilde{r}(s_k, z, a_k) + \gamma \max_{a'} \hat{q}_k(s_{k+1}, z'(s_k, z, a_k), a') - \hat{q}^e_k(s_k, z, a_k)\Big) & \text{if } (s, a) = (s_k, a_k) \\ 0 & \text{otherwise,} \end{cases}$$

where, $z'(s, z, a) := \mathrm{p}(\mathrm{e}(\gamma^{-1}(r(s, a) + z)))$.

Our proposed static CVaR Q-learning can be seen as a special case of the Block-Asynchronous Stochastic Approximation (BASA) studied in (Karandikar & Vidyasagar, 2021). Under an appropriate step size schedule, BASA is guaranteed to converge (Theorem 4.5 and 4.10 in (Karandikar & Vidyasagar, 2021)) to the fixed point of $\bar{\mathfrak{T}}^e$, defined in Appendix B.10.

For completeness, we reproduce the convergence analysis by exploiting the framework presented in Section 4 of (Bertsekas & Tsitsiklis, 1996).

#### B.11.1. CONVERGENCE RESULT IN (BERTSEKAS & TSITSIKLIS, 1996)

Consider some random sequence of iterates $\{\theta_k\}_{k=0}^\infty$, with each $\theta_k \in \mathbb{R}^{|\mathcal{X}|}$ for some finite $\mathcal{X}$, that satisfy:

$$\theta_{k+1}(i) = (1 - \eta_k(i))\theta_k(i) + \eta_k(i)(H\theta_k(i) + \xi_k(i)), \forall i \in \mathcal{X}, \tag{9}$$

where $\xi_k(i)$ is a random noise. Let $\{\mathcal{F}_k\}_{k=0}^\infty$ denote the natural filtration generated by $(\theta_0, \ldots, \theta_k, \xi_0, \ldots, \xi_{k-1}, \eta_0, \ldots, \eta_k)$.

**Assumption B.8** (Assumption 4.3 (Bertsekas & Tsitsiklis, 1996)). The noise terms in (9) satisfy for each $k \geq 0$ and $i \in \mathcal{X}$ that:

(a) $\mathbb{E}[\xi_k(i) \mid \mathcal{F}_k] = 0$ almost surely.

(b) There exist $A, B \in \mathbb{R}$, $\mathbb{E}[(\xi_k(i))^2 \mid \mathcal{F}_k] \leq A + B\|\theta_k\|^2$ almost surely.

**Proposition B.9.** *[Proposition 4.4 (Bertsekas & Tsitsiklis, 1996)] Assume that*

*(a-b) The noise terms $\xi_k$, $k \geq 0$ satisfy Assumption B.8.*

*(c)  For all $i \in \mathcal{X}$, the step-sizes $\eta_k(i)$ are non-negative and satisfy $\sum_{k=0}^\infty \eta_k(i) = \infty$ and $\sum_{k=0}^\infty \eta_k^2(i) < \infty$.*

*(d)  The operator $H$ is a sup-norm contraction with a unique fixed point $\theta^*$.[5]*

*Then, $\theta_k$ converges to $\theta^*$ with probability one.*

#### B.11.2. MAIN PROOF

Let $\mathcal{X} := \mathcal{S} \times \mathcal{Z}_\Delta \times \mathcal{A}$ and define the stacked iterate $\theta_k \equiv \hat{q}^e_k \in \mathbb{R}^{|\mathcal{X}|}$ indexed by $i = (s, z, a) \in \mathcal{X}$ via

$$\theta_k(s, z, a) := \hat{q}^e_k(s, z, a), \qquad (s, z, a) \in \mathcal{X},$$

so that $i$ indexes the flattened Q-table. One can easily verify that Algorithm 2 ensures that the sequence satisfies Eq. (9) with the following definitions:

$$[H\theta](s, z, a) := [\mathfrak{T}^e \theta](s, z, a)$$
$$\xi_k(s, z, a) = \begin{cases} \tilde{r}(s_k, z, a_k) + \gamma \max_{a'} \theta_k(s_{k+1}, z'(s, z, a), a') - [\bar{\mathfrak{T}}^e \theta_k](s_k, z, a_k) & \text{if } (s, a) = (s_k, a_k) \\ 0 & \text{otherwise} \end{cases}$$
$$\eta_k(s, z, a) := \begin{cases} \beta(N_k(s_k, a_k)) & \text{if } (s, a) = (s_k, a_k) \\ 0 & \text{otherwise} \end{cases}$$

Given that $\hat{q}^{*,e}$ is known to be the unique fixed point of the $\gamma$-contraction operator $H = \bar{\mathfrak{T}}^e$ (see Appendix B.10), we are left with verifying the assumptions (a)-(c) in order for the claim made by our theorem to follow from Proposition B.9.

---

[5]Sup-norm contraction is known to imply the weighted sup norm pseudo-contraction assumed in Proposition 4.4 of (Bertsekas & Tsitsiklis, 1996).

**Validating Assumption B.8:** We start by studying the properties of the noise term $\xi_k$. Namely, for all $k \geq 0$ and $(s, z, a) = i \in \mathcal{X}$, one can verify conditions (a) and (b) by considering the two conditions: whether $(s_k, a_k) = (s, a)$ or not. Starting with the easier one:

$$\mathbb{E}[\xi_k(s, z, a) \mid \mathcal{F}_k, (s_k, a_k) \neq (s, a)] = 0$$

and

$$\mathbb{E}\big[\xi_k(s, z, a)^2 \mid \mathcal{F}_k, (s_k, a_k) \neq (s, a)\big] = 0.$$

Alternatively, we have that:

$$\mathbb{E}[\xi_k(s, z, a) \mid \mathcal{F}_k, (s_k, a_k) = (s, a)] = \mathbb{E}_{s' \sim P_{s_k, a_k}}\Big[\tilde{r}(s_k, z, a_k) \ + \ \gamma \max_{a'} \theta_k(s', z'(s, z, a), a') - [\bar{\mathfrak{T}}^e \theta_k](s_k, z, a_k)\Big]$$
$$= [\bar{\mathfrak{T}}^e \theta_k](s_k, z, a_k) - [\bar{\mathfrak{T}}^e \theta_k](s_k, z, a_k) = 0.$$

while one can exploit the fact that for all $(s, z, a) \in \mathcal{X}$, $s' \in \mathcal{S}$, and all $\in \mathcal{L}_\infty(\mathcal{X})$, we have that

$$|\tilde{r}(s, z, a) \ + \ \gamma \max_{a'} \hat{q}(s', z'(s, z, a), a')| \leq |\tilde{r}(s, z, a)| + \gamma |\max_{a'} \hat{q}(s', z'(s, z, a), a')| \leq r_{\max} + \gamma \|\hat{q}\|_\infty.$$

to confirm that:

$$\mathbb{E}\big[\xi_k(s, z, a)^2 \mid \mathcal{F}_k, (s_k, a_k) = (s, a)\big] = \mathbb{E}_{s' \sim P_{s_k, a_k}}\Big[(\tilde{r}(s_k, z, a_k) \ + \ \gamma \max_{a'} \theta_k(s', z'(s, z, a), a') - [\bar{\mathfrak{T}}^e \theta_k](s_k, z, a_k))^2\Big]$$

$$= \mathbb{E}_{s' \sim P_{s_k, a_k}}\Big[(\tilde{r}(s_k, z, a_k) \ + \ \gamma \max_{a'} \theta_k(s', z'(s, z, a), a'))^2\Big] - [\bar{\mathfrak{T}}^e \theta_k](s_k, z, a_k)^2$$

$$\leq \mathbb{E}_{s' \sim P_{s_k, a_k}}\Big[(\tilde{r}(s_k, z, a_k) \ + \ \gamma \max_{a'} \theta_k(s', z'(s, z, a), a'))^2\Big] \leq (r_{\max} + \|\hat{q}\|_\infty)^2 \leq 2 r_{\max}^2 + 2\|\hat{q}\|_\infty^2,$$

where we exploited $(a + b)^2 \leq 2a^2 + 2b^2$.

We thus conclude that:

$$\mathbb{E}[\xi_k(s, z, a) \mid \mathcal{F}_k] = \mathbb{E}[\xi_k(s, z, a) \mid \mathcal{F}_k, (s_k, a_k) = (s, a)] \mathbb{P}\{(s_k, a_k) = (s, a) \mid \mathcal{F}_k\}$$
$$+ \mathbb{E}[\xi_k(s, z, a) \mid \mathcal{F}_k, (s_k, a_k) \neq (s, a)] \mathbb{P}\{(s_k, a_k) \neq (s, a) \mid \mathcal{F}_k\}$$
$$= 0$$

while

$$\mathbb{E}\big[\xi_k(s, z, a)^2 \mid \mathcal{F}_k\big] = \mathbb{E}\big[\xi_k(s, z, a)^2 \mid \mathcal{F}_k, (s_k, a_k) = (s, a)\big] \mathbb{P}\{(s_k, a_k) = (s, a) \mid \mathcal{F}_k\}$$
$$+ \mathbb{E}\big[\xi_k(s, z, a)^2 \mid \mathcal{F}_k, (s_k, a_k) \neq (s, a)\big] \mathbb{P}\{(s_k, a_k) \neq (s, a) \mid \mathcal{F}_k\}$$
$$\leq A + B\|\hat{q}\|_\infty^2,$$

with $A = B = 2$.

**Validating Assumption (c):** Looking more carefully at the random step size sequence $\{\eta_k\}_{k=0}^\infty$, we can confirm that the Robbins–Monro conditions are satisfied. Namely, for all $(s, z, a) \in \mathcal{X}$:

$$\sum_{k=0}^\infty \eta_k(s, z, a) = \sum_{k : (s_k, a_k) = (s, a)}^\infty \beta(N_k(s_k, a_k)) = \sum_{n=1}^\infty \beta(n) = \infty,$$

due to assumptions (1) and (2) of our theorem, whereas

$$\sum_{k=0}^\infty \eta_k(s, z, a)^2 = \sum_{k : (s_k, a_k) = (s, a)}^\infty \beta(N_k(s_k, a_k))^2 = \sum_{n=1}^\infty \beta(n)^2 < \infty,$$

due to the same assumptions.

This completes our proof. $\square$

## C. Algorithms

---

**Algorithm 3** Outer Optimization for Static CVaR

---

**Require:** Learned tabular $\hat{q}^e(s, z, a)$ on $\mathcal{Z}_\Delta$, risk level $\alpha \in [0, 1]$, initial state $\bar{s}_0 \in \mathcal{S}$.

1: **Initialize:** $\hat{\psi}_\alpha(\bar{s}_0) \leftarrow -\infty, \quad z_\alpha^* \leftarrow 0$
2: Compute greedy policy: $\tilde{\pi}(s, z) \leftarrow \arg\max_{a \in \mathcal{A}} \hat{q}^e(s, z, a)$
3: **for** each $\tilde{z} \in \mathcal{Z}_\Delta$ **do**
4:   $\hat{J}(\tilde{z}) \leftarrow -\tilde{z} + \frac{1}{\alpha}\Big( -\tilde{z}_- + \max_{a \in \mathcal{A}} \hat{q}^e(\bar{s}_0, \tilde{z}, a) \Big)$
5:   **if** $\hat{J}(\tilde{z}) > \hat{\psi}_\alpha(\bar{s}_0)$ **then**
6:    $\hat{\psi}_\alpha(\bar{s}_0) \leftarrow \hat{J}(\tilde{z})$
7:    $z_\alpha^* \leftarrow \tilde{z}$
8:   **end if**
9: **end for**
10: **Return** Optimal budget $z_\alpha^*$, CVaR value $\hat{\psi}_\alpha(\bar{s}_0)$, and optimal policy $\tilde{\pi}(s, z)$

---

**Algorithm 4** Static CVaR Policy Execution in Nominal MDP

---

**Require:** Risk level $\alpha \in [0, 1]$, corresponding optimal initial budget $z_\alpha^*$, optimal policy $\tilde{\pi}(s, z)$, discount $\gamma \in (0, 1)$, maps p and e, nominal MDP $\mathcal{M}$.

1: $s \leftarrow \bar{s}_0, \quad z \leftarrow z_\alpha^*$
2: **while** $s$ is not terminal **do**
3:   $a \leftarrow \tilde{\pi}(s, z)$
4:   Execute $a$ in $\mathcal{M}$ and observe $s' \sim P_{s,a}$, reward $r$
5:   Update augmented state: $z' \leftarrow \text{p} \circ \text{e}\big(\gamma^{-1}(r + z)\big)$
6:   $s \leftarrow s', \quad z \leftarrow z'$
7: **end while**

---

## D. Experiment Details

### D.1. Crater Walk Environment

To empirically test our proposed static CVaR value iteration and Q-learning algorithms, we use a $4 \times 5$ stochastic grid world environment, where the goal is for an autonomous robot to safely (without falling into the crater) travel from a starting position "S" to the goal position "G" while being fuel efficient.

The robot can move in four directions (forward, backward, left, right). Due to the sensor and actuator noise, the probability that the robot moves in the intended direction is $1 - \omega$, to either left or right of its current position with a $\frac{4\omega}{9}$ probability and backwards with probability of $\frac{\omega}{9}$. If the robot tries to move into the environment wall, it remains in its current position with probability $1 - \omega$. The goal state is absorbing, i.e. once the robot enters the goal, it remains there in perpetuity, acquiring zero reward. For our experiment, we set $\omega := 0.25$.

At each stage, the robot receives a penalty of $-1$ to indicate the fuel usage. There is a crater that the robot should avoid entering, indicated by gray cells. To escape the crater (move to non-gray positions), the robot needs extra fuel which consequently yields a penalty of $-10$. The goal state is absorbing, with a reward of 0. The discount factor is set as $\gamma := 0.9$.

### D.2. Discretizing Augmented State

To test our algorithms in a tabular setting, we first discretize the augmented state. Based on the environment's reward model, the range of the infinite-horizon discounted return, consequently the range of the continuous augmented state is in $[-10(1-\gamma)^{-1}, 10(1-\gamma)^{-1}]$. Consequently, $\mathcal{Z}_\Delta$ is defined by a uniform grid on the range $[-100 - \nu, 100 + \nu]$, with 5000 bins. We set $\nu := 10$. The projection map p is implemented using the `clip` function where any value outside the $z$ domain is clipped to the nearest domain limit. The rounding maps $\{l_\Delta, u_\Delta\}$ are implemented using `floor` and `ceil` functions respectively.

## D.3. Static CVaR Value Iteration

We run the static CVaR Value Iteration (Algorithm 1) using 10 independent seeds. The threshold tolerance for stopping the algorithm is set to $\delta := 1e - 4$. The algorithm estimates the optimal action-value function $\hat{q}^{*,e}$ which is then used to compute the optimal starting budgets $z_\alpha^*$ for 12 different risk levels $\alpha \in \{0.0, 0.01, 0.05, 0.1, 0.2, 0.3, 0.4, 0.5, 0.6, 0.7, 0.8, 0.9, 1.0\}$ by running the outer optimization (Algorithm 3). Finally, we can deploy the optimal policy and test the behavior of the robot in the environment using the policy execution described in Algorithm 4. We run the policy for each $\alpha$ and collect 10k trajectories. These are used to calculate the empirical return distribution, the empirical CVaR performance as well as to analyze the number of times the robot enters the crater per episode.

## D.4. Static CVaR Q-learning

The static CVaR Q-learning algorithm (Algorithm 2) is also run 10 times, with different seeds. The algorithm is trained with $M := 75k$ episodes where an episode is considered a successful entry into the goal state. At the end of each episode the environment is reset. During training, the agent is reset to any of the safe positions in the grid. However, at evaluation, the policy is tested with the initial state "S". We employ a $\varepsilon-$greedy policy where $\varepsilon$ controls the explorations. The initial value of $\varepsilon := 1$ and it is gradually decreased until $\varepsilon := 0.1$ using a linear decay schedule. The step sizes are determined by the state-action visitation count. Unless stated otherwise in the figures, we use 5000 bins to discretize the augmented state space. During training, the Q-update step sizes are computed as:

$$\beta_k(s, a) = \max\left(\kappa_{\min}, \frac{\kappa}{1 + \lambda N_k(s, a)}\right),$$

where $N_k(s, a)$ represents the (state, action) visit count up to step $k$. The hyper-parameters used for running experiments are listed in Table 2. We will release the implementation of both our algorithms upon paper acceptance.

*Table 2.* Hyper-parameters used for running Q-learning experiments.

| Training setup | Test setup | Q-learning hyper-parameters |
|---|---|---|
| Episodes $M = 75,000$ | Number of rollouts: 10,000 
 Max. environment steps: 150 | Discretizations bins = 5,000 
 $\kappa = 1.0$ 
 $\kappa_{min} = 10^{-4}$ 
 $\lambda = 0.01$ 
 $\varepsilon_{\text{start}} = 1.0, \varepsilon_{\text{end}} = 0.1$ 
 Total decay steps for $\varepsilon$: $10^8$ |

