# OpenReview forum: "Reward Redistribution for CVaR MDPs using a Bellman Operator on L-infinity"
_ICML.cc/2026/Conference — ICML 2026 spotlight_

### Official Review · Reviewer_YkW3 · 2026-03-08

**Soundness:** 4
**Presentation:** 3
**Significance:** 3
**Originality:** 4
**Overall Recommendation:** 5
**Confidence:** 4

**Summary:**

In this paper, the authors proposed an alternative formulation of reinforcement learning with risk-sensitive objective. Consequently, new bellman operator with improved properties (contraction property on a larger function space and more informative propagation) are obtained and analyzed, new algorithms for learning with continuous augmented variables are proposed and tested.

**Compliance With Llm Reviewing Policy:**

Affirmed.

**Final Justification:**

The rebuttal addresses my concerns, and I can also see that while more extensive numerical experiments are certainly desirable, the example presented in the paper is representative and carefully design. I thus raise my score to five.

**Key Questions For Authors:**

1) The constant $K$ in discretization granularity $\Delta=r_\gamma/K$ appears to be arbitrary, does its selection affect the performance of the algorithm?

2) If the non-positive reward assumption 5.1 could be easily removed as claimed in the footnote, would it be better to just remove it and proceed as the footnote suggested?

**Limitations:**

Yes

**Strengths And Weaknesses:**

Strength:

The theoretical results are vey interesting, the improvement from Bauerle & Ott is subtle and the benefit is obvious. The discussions on the discretization is also clear and effective.


Weakness:


1) The experiments are rather weak. Only one 2D problem of moderate size has been experimented on. More extensive tests and evaluations are definitely needed.

2) Information on the reference "Predictive CVaR Q-learning" should be updated.

3) The exposition in the last paragraph of Page 4 can certainly be improved.

---

> ### Author Rebuttal · Authors · 2026-03-29
>
> Thank you for your constructive feedback and for recognizing the originality of our work, the improvements over Bäuerle \& Ott (2011), and the clarity of our discretization discussions. We address your specific questions and concerns below.
>
> **Answers to the Author Questions:**
> 1. **Discretization constant $K$:** The constant $K$ used to discretize the augmented state $z$ is chosen to balance the approximation quality against computational cost. In particular, the choice of $K$ sets the discretization granularity $\Delta := r_\gamma/K$, a larger $K$ produces a finer discretization and tighter approximation guarantees, at the expense of more expensive dynamic programming over the enlarged state space. This trade-off is made precise in Theorem 5.3 and Theorem 5.4, the latter provides upper and lower bounds on the gap between the CVaR performance  $\Psi^*$ under the true optimal policy vs the CVaR performance of the approximate solution. Figure 4 also illustrates the effect of discretization resolution on CVaR performance.
> 2. **Non-positive Reward Assumption:** Thank you for raising this point. In the discretization analysis, the one-sided reward assumption is needed so that the upper- and lower-bounding Bellman operators for the finite augmented MDP preserve their ordering, i.e., the monotonicity used throughout the proofs (Proposition 5.3, Theorem 5.3 and 5.4). We chose to present the non-positive case because risk-sensitive RL commonly models risky states and actions through penalties, making this setting especially natural. That said, the same analysis extends to the non-negative case as well, and the corresponding modifications are indicated in Footnote 3. Moreover, due to the translation invariance property of CVaR, one could also replace $r(s,a)$ with $r(s,a)-r_\gamma$ to obtain a new MDP with non-positive reward while preserving the same optimal CVaR policy. We can make this point more explicit in the revision, however, we prefer keeping the non-positive reward assumption as is, mainly for the simpler presentation.
>
> **Addressing the Weakness:**
> 1. **Weak Experiments:** We thank the reviewer for this comment. Our main contribution is to rigorously ground static CVaR problems in a recursive dynamic programming framework and establish the exact theoretical foundation underlying the proposed new Bellman operator. Accordingly, the experiments are designed to validate these theoretical claims in a controlled setting. Although moderate in size, the chosen environment captures the key features needed to evaluate risk-averse behavior, including inherent stochasticity and risky states that induce changes in the optimal policy as the level of risk aversion varies. Its penalty-based reward structure also naturally aligns with the non-positive reward setting analyzed in our framework. We therefore believe the current experiments are appropriate for the scope of the paper and effectively illustrate the practical value of the proposed CVaR decomposition.
> 2. **Citation Update:** The "Predictive CVaR Q-learning" paper was still under double-blind review at the time of our submission. We now have the correct, de-anonymized citation and will update the references in our final version accordingly.
> 3. **Page 4 Exposition:** Thank you for pointing this out. We have rewritten the last paragraph on Page 4 to improve readability. Here is the updated version: "Even when the original state space is tabular, the augmented value function remains challenging to handle because its second argument, $z \in [-r_\gamma, r_\gamma]$, is continuous. In the following result we show that $\bar{v}^\ast$ can be approximated accurately using a uniform discrete grid over $z$. This is made possible by two key properties: the superadditivity of $(x+y)_{-}$ and the fact that $\bar{v}^\ast$ is Lipschitz continuous in $z$."
>
> We hope these answers adequately address your questions and help clarify the significance and intended scope of our contribution.

---

> > ### Author Rebuttal · Reviewer_YkW3 · 2026-04-01
> >
> > The rebuttal addresses my concerns, and I can also see that while more extensive numerical experiments are certainly desirable, the example presented in the paper is representative and carefully design. I thus raise my score to five.

---

### Official Review · Reviewer_NYVw · 2026-03-12

**Soundness:** 3
**Presentation:** 3
**Significance:** 4
**Originality:** 3
**Overall Recommendation:** 5
**Confidence:** 3

**Summary:**

The paper studies static CVaR optimization in infinite-horizon discounted MDPs. Building on the augmentation idea of Bauerle & Ott, the papper propose an algebraic reformulation that produces a transformed Bellman operator (they also prove it to be a contraction operator) on the augmented state  $(s,z)$. Through their literature review, the authors bring to notice that a Cvar bellman previously obtained is in fact erroneous. Building on works that attempt to fix this, the authors provide meaningful contribution by in some sense closing the lid on an open question.
Using the contraction property,  the authors provide a risk-sensitive value iteration algorithm and in settings where the transition probability matrix in unknown, provide a  Q-learning algorihtm. Thye provide empirical results on a stochastic gridworld problem.

**Compliance With Llm Reviewing Policy:**

Affirmed.

**Key Questions For Authors:**

none

**Limitations:**

yes

**Strengths And Weaknesses:**

The paper makes an important theoretical fix to a known practical problem in static CVaR MDPs by designing an operator that is both exact for the CVaR objective and is also a contraction. The related work is comprehensive and motivation is clearly explained. The paper is also well written.  The new state augmentation is novel and interesting. The theoretical results appear correct to me and the contributions certainly are useful.

Some weak aspects are as follows
The rigorous analysis and experiments are confined to tabular MDPs and uniform discretization of z. The paper states that the operator extends to function approximation, but no theory or experiments i think are provided that explore approximation error with parametric function approximators (neural nets.

---

> ### Author Rebuttal · Authors · 2026-03-29
>
> Thank you for your careful reading and positive assessment of the paper. We appreciate your recognition that the paper makes an important theoretical contributions to static CVaR RL, as well as your comments on the clarity of the motivation, related work, and presentation. We also agree that extending the proposed operator to parametric approximators and deep RL methods is an important direction. We view our current work as establishing a rigorous theoretical foundation for a new static CVaR Bellman operator. For this, a tabular setting serves as the cleanest setting for validating the CVaR reformulation, contraction analysis, and discretization guarantees, and we leave extensions to function approximation as a natural direction for future work.

---

> > ### Author Rebuttal · Reviewer_NYVw · 2026-04-01
> >
> > I had no major concerns to be addressed.

---

### Official Review · Reviewer_ZZM1 · 2026-03-13

**Soundness:** 4
**Presentation:** 4
**Significance:** 3
**Originality:** 3
**Overall Recommendation:** 5
**Confidence:** 4

**Summary:**

This paper is about risk-sensitive criteria in control. It proposes a different formulation of the problem for the CVaR criterion, which does away with the unpleasant time-inconsistency problems which usually preclude dynamic programming in this framework. It does this via state-augmentation, modifying prior work by Bäuerle & Ott (2011) through an exact reformulation of the objective. A Bellman operator is then constructed for this reformulated MDP, which is contractive on the space of bounded functions and can thus be used to get a fixed point in the usual manner. Numerically, the only issue remaining is the fact that the augmented part of the state-space is continuous, which is addressed through discretisation bounds.

Finally, the paper showcases the formal development of a value iteration algorithm and a Q-learning algorithm, with guarantees. Of note, because the augmented MDP is deterministic, Q-learning can be modified to update the Q-values for all deterministic parts of the state space from each sample, leading to efficiency gains.

**Compliance With Llm Reviewing Policy:**

Affirmed.

**Key Questions For Authors:**

No major questions.

**Limitations:**

Yes

**Strengths And Weaknesses:**

### Soundness
The paper is mostly theoretical, and in this regard it is very sound: arguments are rigorous, well presented and easy to follow. Theorems are clearly stated mathematically and clearly contextualised and explained in the surrounding text.

In terms of experiments, there is only one environment presented. That being said, as the paper is theoretical, I think the experiments are better thought of as an illustration, and in that regard, the choice of this one environment is sound, as it presents a discontinuous behaviour in the threshold probability. Description of the experimental set-up (number of runs, seeds, etc.) was also clear.

### Presentation
The paper is very well written. The prose is crisp and to the point, delivering exactly the information required and telling an engaging story. In particular, the lengthy presentation of risk-sensitive RL is very precise and, I think, a good onboarding for people from RL who haven’t been exposed to it before, which I think will increase the reach of the paper.

Minor comments:
- typos l113 left, l336 right, l831
- Figure 3d could be on a log-log scale for better readability (provided the data isn’t too noisy in the tail)

### Significance
Risk-sensitive RL is an up-and-coming field, garnering significant interest. However, results for the static problem have often been unsatisfactory to me. In contrast, this paper offers a clear and rigorous reduction, not only to another MDP but to a finite MDP (up to approximation bounds they provide). This result provides a basis, not only for VI and Q-learning but also for any other control/RL algorithm, which I think is a great step forward, at least for the CVaR objective. While the core trick on which the analysis rests is very simple, they deserve credit for spotting something overlooked, especially if it is a simple trick.

### Originality
Risk-sensitive RL is, in my opinion, still quite an original and fresh research direction (at least for RL theory). While I’m not active in this field, clever modification of the original static problem to obtain a working DPP seems quite original to me.

---

> ### Author Rebuttal · Authors · 2026-03-29
>
> Thank you for your encouraging feedback, your careful reading of our paper, and your strong support. We are glad that you found the manuscript well-written, theoretically sound, and a great step forward for the risk-sensitive RL community. Thank you for noting that the paper’s primary contribution is theoretical, with experiments designed to validate our theoretical claims. We appreciate your helpful feedback regarding the typographical errors and the presentation of Figure 3(d), and we will carefully take your suggestions into account in preparing the final version.

---

> > ### Author Rebuttal · Reviewer_ZZM1 · 2026-04-05
> >
> > No concerns.

---

### Decision · Program_Chairs · 2026-04-30

**Decision:**

Accept (spotlight)

**Comment:**

Across reviewers, the acceptance recommendation rests on a clear conceptual advance: the paper identifies a subtle but impactful reformulation of the static CVaR objective that restores a valid dynamic programming structure. As highlighted by ZZM1 and NYVw, this state-augmentation and exact objective reformulation resolves a long-standing issue of time-inconsistency while yielding a contractive Bellman operator—a property that is both nontrivial and foundational for algorithm design. Importantly, the contribution is not merely algorithmic but structural: it provides a clean reduction of a difficult risk-sensitive problem to a standard MDP framework, enabling the direct application of value iteration, Q-learning, and other RL tools. Reviewers consistently note that while the underlying “trick” is simple in hindsight, its identification is non-obvious and closes a gap in the literature. The resulting proofs—particularly the contraction and discretization guarantees—are seen as technically sound and conceptually clarifying, offering a unifying lens that prior work lacked. Even reviewers who noted limitations (e.g., tabular scope, modest experiments) viewed these as secondary to the theoretical contribution, which they judged to be original, rigorous, and broadly enabling for future work in risk-sensitive RL. In sum, the novelty lies not in complexity but in the precision of the reformulation and the resulting proof structure, which together establish a principled and extensible foundation for CVaR-based control.